# Metal-responsive regulation of enzyme catalysis using genetically encoded chemical switches

Yasmine S. Zubi [1,4], Kosuke Seki[2,4], Ying Li [3], Andrew C. Hunt [2], Bingqing Liu[1], Benoît Roux [3✉], Michael C. Jewett [2✉] & Jared C. Lewis [1✉]

Dynamic control over protein function is a central challenge in synthetic biology. To address this challenge, we describe the development of an integrated computational and experimental workflow to incorporate a metal-responsive chemical switch into proteins. Pairs of bipyridinylalanine (BpyAla) residues are genetically encoded into two structurally distinct enzymes, a serine protease and firefly luciferase, so that metal coordination biases the conformations of these enzymes, leading to reversible control of activity. Computational analysis and molecular dynamics simulations are used to rationally guide BpyAla placement, significantly reducing experimental workload, and cell-free protein synthesis coupled with high-throughput experimentation enable rapid prototyping of variants. Ultimately, this strategy yields enzymes with a robust 20-fold dynamic range in response to divalent metal salts over 24 on/off switches, demonstrating the potential of this approach. We envision that this strategy of genetically encoding chemical switches into enzymes will complement other protein engineering and synthetic biology efforts, enabling new opportunities for applications where precise regulation of protein function is critical.

[1] Department of Chemistry, Indiana University, Bloomington, Indiana, USA. [2] Department of Chemical and Biological Engineering and Center for Synthetic Biology, Northwestern University, Evanston, IL, USA. [3] Department of Biochemistry and Molecular Biology, University of Chicago, Chicago, IL, USA. [4]These authors contributed equally: Yasmine S. Zubi, Kosuke Seki. ✉email: roux@uchicago.edu; m-jewett@northwestern.edu; jcl3@iu.edu

Naturally occurring enzymes have evolved to catalyze chemical reactions, in many cases, with exquisite selectivity, substrate specificity, and high catalytic rates[1]. Many enzymes also possess regulation and control mechanisms to ensure that they can respond to environmental stimuli[2]. Leveraging these remarkable properties for chemical synthesis has long appealed to chemists and synthetic biologists[3,4], but current enzyme engineering strategies largely focus on improving catalytic properties or changing substrate specificity[5]. These engineering strategies often neglect native control capabilities like allosteric regulation, which can be inefficient for non-native substrates[6], or lost without selection pressure as has been observed in several directed evolution approaches[7–10]. Most engineered enzymes are used in highly optimized single-reaction processes[11,12] where allosteric control is not needed and could even be detrimental. The need for modular, orthogonal methods to control enzyme activity in a stimulus-dependent manner, however, has become increasingly apparent[13] for efforts to evolve proteins for in vitro multi-enzyme cascade catalysis[3], diagnostic tools[14–16], biosensing[17,18], and construction of novel signaling circuits[19].

A key challenge to engineering dynamically regulated enzymes is the complex interplay between catalytic activity, substrate specificity, and regulation. For example, while native allosteric proteins have been engineered to respond to new stimuli, this control typically applies to the native protein function, frequently DNA binding[6,20]. Chimeric systems have also been developed to couple a naturally responsive protein to a protein of interest (POI) so that regulation of the former allows for control over the latter[21]. Optical regulation is particularly notable in this regard given that it allows for precise spatiotemporal control of protein function[22]. This approach is commonly achieved by fusing a POI to the light-oxygen-voltage (LOV) responsive domain of plant phototropins, which can control protein function through blue-light induced conformational changes (e.g., sterically hindering accessibility to enzyme active sites or control ordered and disordered protein regions)[23]. Optogenetic dimerization has been successfully implemented in systems that utilize LOV domains[24,25] and those based on *Arabidopsis thaliana* cryptochrome 2 (CRY2)[26]. These approaches often require that large (potentially disruptive) regulatory domains be used[27,28], and achieving efficient regulatory transduction between domains can require extensive protein engineering[29]. Finally, de novo design has been used to produce switchable proteins[30,31], but switching has only been reported in response to peptides and proteins, and de novo design of switchable enzymes has not been reported.

We envisioned that small molecule switches that undergo reversible formation or cleavage of covalent bonds in the presence of different stimuli could be integrated into enzymes to enable regulation of catalysis. For example, boronic acids and diols form boronic esters at low pH[32], hydrazines and aldehydes form hydrazones in the presence of anilines[33], and bipyridines bind metal ions to form bis- or tris-bipyridine complexes[34]. We reasoned that amber codon suppression methods[35] could be used to genetically encode non-canonical amino acids (ncAAs) that contain the reactive functional groups (linking groups, LGs) found in synthetic switches. Previous studies have established that ncAAs can be integrated into proteins and peptides to reversibly control conformation[36]. Recently it has also been shown that ncAAs can be used to control protein function[37,38], although most examples rely on irreversible mechanisms like deprotection of caged residues[39–41]. Reversible control of protein function has been demonstrated by incorporating light-responsive ncAAs, such as those based on azobenzenes, into an enzyme that is already subject to allosteric regulation[41,42]. Key to our approach, however, is the use of LGs to control the conformation of proteins that are not already subject to allosteric control to enable reversible control over enzyme catalysis. LG incorporation would involve minimal disruption of protein structure and would only require that the LGs can be placed in an orientation to disrupt catalysis upon metal binding. The structural simplicity of the LG approach, however, belies the potential difficulty of engineering systems in which two LGs impart suitable switching properties.

To address this difficulty, here we developed a framework to design, build, test, and analyze enzymes with genetically encoded chemical switches. Initial efforts focused on bipyridine (Bpy) LGs that could induce reversible activation of enzymes in the presence/absence of metal salts (Fig. 1a). Specifically, bis-bidentate metal binding by the Bpy LGs would restrict the enzyme to a closed conformation incapable of turnover while removal of the metal would allow the enzyme to access conformational changes required for turnover. We envisioned that this mode of regulation could be used in vitro to cycle enzyme activity or under a broader range of situations where a single metal-dependent activation or deactivation is needed. Our framework (Fig. 1b) used molecular dynamics (MD) simulations to identify all residue pairs that undergo significant changes in $C_\beta$-$C_\beta$ distance between enzyme conformations. Enzyme variants containing LG pairs at candidate sites were then designed and rapidly prototyped using cell-free protein synthesis (CFPS)[43] and high-throughput experimentation (HTE) to select active enzyme variants with metal-dependent activity for analysis. Metal-responsive variants[44] of two distinct enzymes not subject to native allosteric control were thus engineered by disrupting catalytically relevant conformational dynamics via LG-metal binding. This approach complements existing methods for tuning enzyme catalysis and providing a means to genetically encode functionally orthogonal control elements to regulate enzyme catalysis.

## Results

**Model selection and design of conformational switches.** Allosteric modulation of enzyme activity requires an enzyme that can adopt at least two states possessing different activity[2]. By perturbing the populations of these states via covalent bond formation and cleavage under different conditions, LGs could be used to regulate enzyme activity in a stimulus-responsive manner. We selected prolyl oligopeptidase (POP), from the hyperthermophilic organism *Pyrococcus furiosus* (*Pfu*) as a model test case for LG regulation of enzyme activity. POP is a serine protease that hydrolyzes small peptide substrates containing proline residues[45], and it is not subject to native allosteric regulation. The POP structure comprises a β-propeller domain that controls substrate access to the active site and a peptidase domain that contains a Ser-His-Asp catalytic triad. MD simulations revealed that POP undergoes spontaneous domain opening/closing that provides a dynamic port of entry for substrates and orients the catalytic triad His residue into a catalytically competent position[46]. Suitably placed LGs could therefore provide a means to control POP conformation and peptidase activity.

POP is a 72 kDa protein with 189,420 possible double variants. As many proteins of interest could have a similarly large number of potential LG site pairs, or more, we developed a computationally guided approach for identifying sites for LG incorporation that could be applied to any protein that undergoes relevant conformational domain changes. Structural analysis and MD simulations were used to identify residues that (i) are close enough in space to enable covalent bond formation between LG pairs in one enzyme conformational state (e.g., the closed conformation of POP), (ii) are solvent exposed to allow the proper chemistry to take place, (iii) undergo suitable changes in $C_\beta$-$C_\beta$ distance to prohibit bond formation in a different

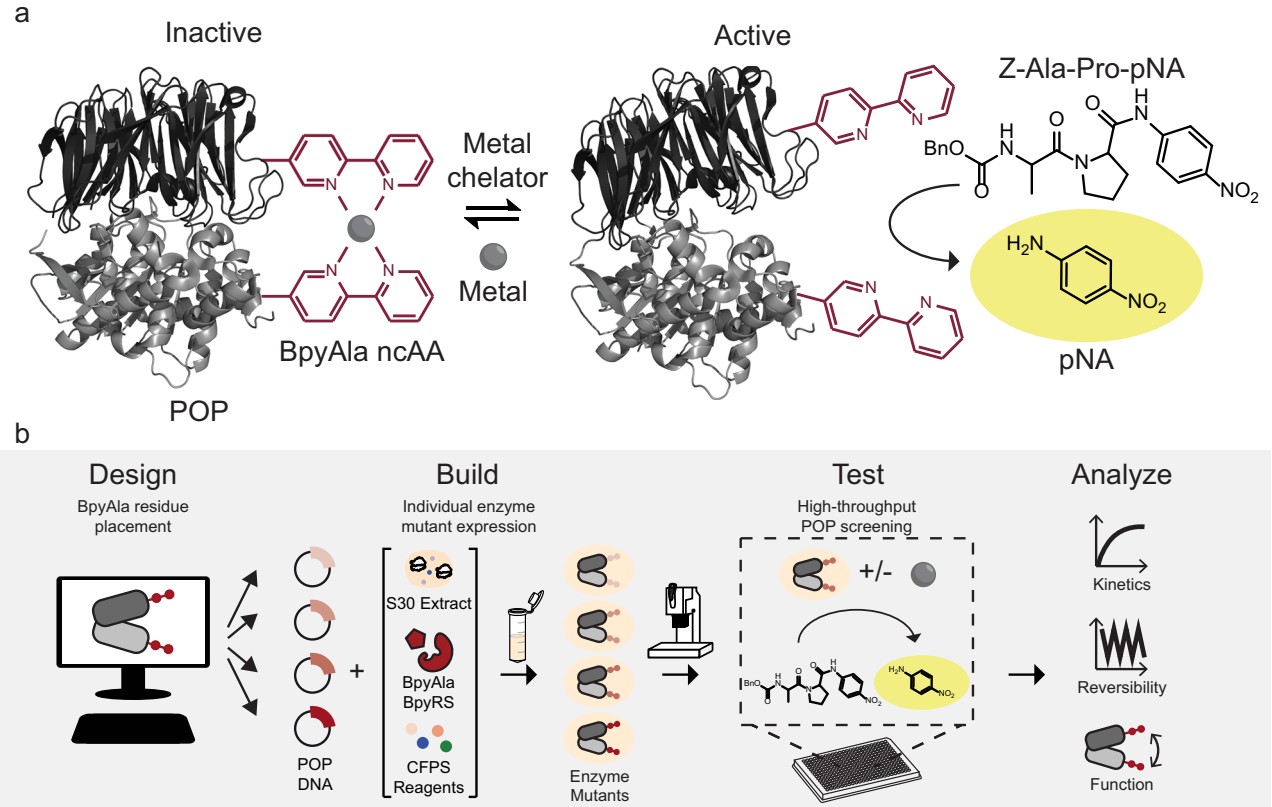

**Fig. 1 Development of BpyAla LG protein switches. a** Reversible metal regulation of *Pfu* prolyl oligopeptidase (POP) via bis-bidentate metal binding by a BpyAla LG pair. **b** Workflow for engineering LG protein switches.

conformational state (e.g., the open conformation of POP), and (iv) are placed on opposite domains of the enzyme (Fig. 2a). Because we were targeting metal regulation using genetically encoded bipyridinylalanine (BpyAla) residues[47], the structure of a Zn(Bpy)₂ complex was analyzed to establish that a distance of approximately 10.5 Å would allow for formation of a metal bis-BpyAla complex, M(BpyAla)₂, at the interface of the β-propeller and peptidase domains of POP (Fig. 2b). Pairs of residues with distances from 9.5-11.5 Å between $C_β$ were selected and ranked according to the change in $C_β$-$C_β$ distances observed during the simulations (Fig. 2c). As a result, 27 LG pairs were selected for evaluation.

**HTE screening of POP switches**. We next developed a three-step, HTE approach to synthesize and screen metal ion regulation of all computationally designed POP variants (Fig. 1b). First, we developed an *E. coli*-based CFPS platform[43,48–50] to synthesize proteins containing two BpyAla residues in response to the amber stop codon. BpyAla and aminoacyl-tRNA synthetase (BpyRS)[47] concentrations were combinatorially optimized using a sfGFP reporter containing two premature amber stop codons (2TAG-sfGFP) to enable 2TAG-sfGFP expression at yields comparable to wild-type (WT) sfGFP, and BpyAla incorporation was verified by intact protein ESI-MS (Supplementary Figure 1). Second, we synthesized the entire panel of POP variants (POP$_{X/Y}$, where X and Y are the sites of BpyAla incorporation) at an average yield of 1098 ± 124 μg/mL within a 15 μL CFPS reaction, comparable to WT protein (Supplementary Figure 2). All POP variants were synthesized at full-length as measured by autoradiography of ¹⁴C-labeled proteins with minimal truncation products, and accurate incorporation of the BpyAla pair was further confirmed by intact protein ESI-MS (Supplementary Figure 2). Third, we

designed a screening workflow that involved high-throughput liquid-handling to set up 384-well plate-based enzyme assays using POP-enriched CFPS reactions (diluted to be equimolar using blank CFPS reactions). POP activity and metal responsiveness were measured spectrophotometrically by monitoring *p*-nitroaniline (pNA) released by enzymatic hydrolysis of Z-Ala-Pro-pNA (Fig. 1a)[45]. The HTE workflow is purification-free and could allow for analysis of hundreds of reactions in less than a day, with the potential to screen large numbers of variants with a suitable functional assay.

We first used this workflow to screen POP variants for Ni(II) inhibition in the presence of 0–1,000-fold molar excess of Ni(II) relative to enzyme (Fig. 3a, 0.1 μM enzyme). Control reactions indicated that blank CFPS extracts and 2TAG-sfGFP were inactive, and no background inhibition of WT POP occurred at any Ni(II) concentration. On the other hand, all but five POP variants displayed activity, and ten showed enzymatic rates >50% that of WT. Of these ten, six showed dose-dependent inhibition by Ni(II), which we defined as less than 50% apo activity at the highest concentration of metal tested (Supplementary Table 1). The top four variants with the greatest degree of inhibition, POP$_{167/517}$, POP$_{169/510}$, POP$_{159/517}$, and POP$_{169/512}$, displayed nearly complete inhibition (≤5% apo activity) at higher concentrations of Ni(II) while maintaining >75% of WT activity in the absence of metal (Fig. 3a, shaded graphs, red text). At intermediate concentrations of Ni(II), these four variants exhibited 2.8-14 fold decreased rates. Interestingly, POP$_{167/513}$ displayed >50% increase in activity in the presence of Ni(II) relative to the apo protein at the highest concentration of metal tested (Fig. 3a, shaded graph, blue text). The proximity of 167/517 and 167/513 in primary sequence space highlights how subtle changes in protein structure can have significant impacts on function (i.e., metal inhibition and activation, respectively) and

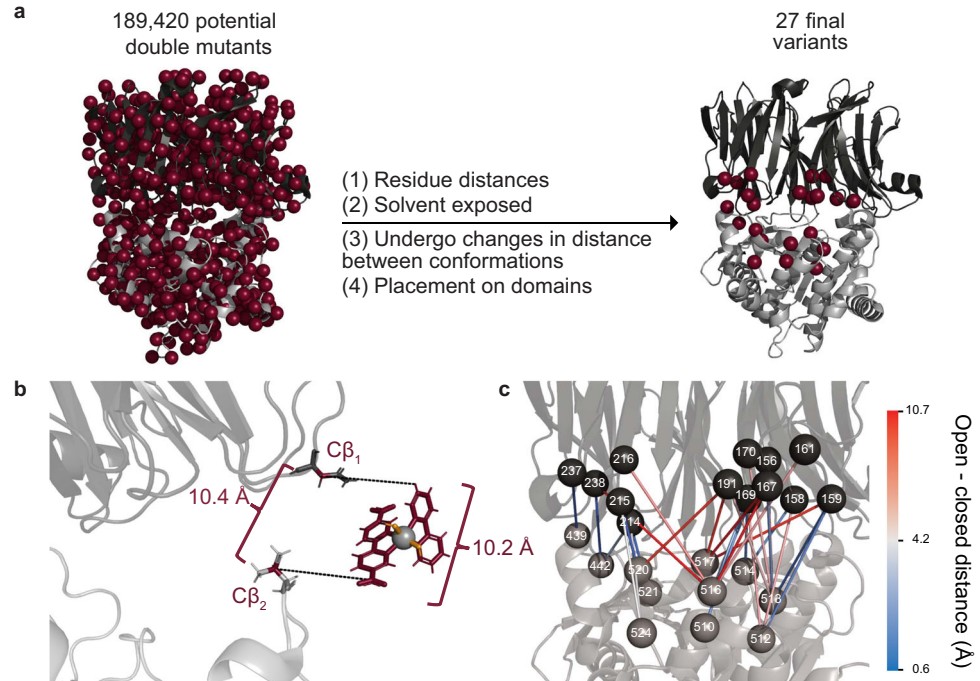

**Fig. 2 Design strategy for Bpy-Ala POP variants. a** Filtering process used to select sites for Bpy LG pairs. **b** Distance and geometry requirements for Bpy LG placement were obtained by mapping the structure of a Zn(Bpy)$_2$ (CCDC:756656) complex onto the *Pfu* POP structure (PDB ID: 5T88). A distance of 10.5 ± 0.5 Å between the C-5 substituents on the two Bpy ligands and the Cβ atoms of the selected residues was used to ensure that Zn(BpyAla)$_2$ complex could be accommodated between the residues. **c** Locations of the final 27 LG pairs. The spheres represent the positions of the Cβ atoms of the selected residues on the β-propeller (black) and peptidase (gray) domains and labels represent residue numbers. Lines are drawn between residues in each pair, and the color of the line represents the change in distance between the residues in the open and closed POP conformations. Source data are provided as a Source Data file.

the relatively unexplored space in understanding conformational effects on protein function.

We next characterized POP$_{X/Y}$ responses to different divalent metal cations, including Cu(II), Co(II), Zn(II), and Fe(II) (Fig. 3b and Supplementary Figure 3). While WT POP did not respond to these metals, POP$_{169/510}$, POP$_{159/517}$, POP$_{167/517}$, and POP$_{169/512}$ were inhibited by Cu(II), Co(II), and Zn(II) in a dose-dependent manner, but only the latter two were inhibited by Fe(II). POP$_{167/513}$, which was activated by Ni(II), was also activated by Cu(II) and Co(II) but not Zn(II) or Fe(II) (Fig. 3b, Supplemental Figure 3). These differences in extent of activation and inhibition may reflect different preferred coordination geometries of the M(Bpy)$_n$ complexes or the relative binding affinities of the Bpy LG for the different metals[51].

The reversibility of POP variant inhibition/activation was examined by adding a competitive chelator, ethylenediaminetetraacetic acid (EDTA), to reactions conducted using the workflow outlined above (Fig. 3b, Supplementary Figure 4). EDTA formation constants for complexation with Ni(II), Cu(II), Co(II), Zn(II), and Fe(II) are ~10$^{14}$–10$^{18}$ (Fe < Co < Zn < Ni < Cu), which exceed values for M(Bpy)$_2$ complex formation and thus should enable reversibility by competition[52]. All POP variants were incubated with 1000-fold excess metal, exposed to 2 mM EDTA, and assayed to measure activity. Nearly all enzymes tested with Ni(II), Cu(II), Co(II), or Zn(II) displayed near-quantitative recovery of activity upon addition of EDTA (Fig. 3b). Recovery of activity was not observed following addition of EDTA to variants treated with Fe(II), perhaps due to the relatively low affinity of EDTA for Fe(II) relative to the other metals tested. Activation of POP$_{167/513}$ was also found to be reversible by EDTA-based chelation. These results highlight how our HTE workflow facilitated rapid identification of POP variants that displayed

reversible catalytic responses to divalent metals in a manner consistent with the formation of a M(BpyAla)$_2$ complex involving Bpy LGs.

**Functional characterization of a BpyAla POP switch**. We next studied the catalytic properties of POP$_{167/517}$, POP$_{169/510}$, POP$_{159/517}$, POP$_{169/512}$ and POP$_{167/513}$, which exhibited the highest degrees of reversible inhibition or activation by metal salts in our HTE workflow, by analyzing initial rates (Fig. 3c, Supplementary Figure 5). Purified variants were produced in yields of approximately 50 mg/L, and BpyAla incorporation was confirmed by intact protein ESI-MS. The isolated variants included ~5–40% of POP that had been truncated at the second amber stop codon due to the inherent affinity of POP for Ni-NTA resin, though >90% purity was observed for POP$_{167/517}$, POP$_{167}$, and POP$_{517}$ (Supplementary Fig. 6). Steady state kinetic parameters measured at 85 °C for each variant (Table 1, Supplementary Table 2) qualitatively matched the activity trends observed in the high throughput screen. Particularly notable is POP$_{167/517}$, which displayed a 17-fold change in activity in response to Ni(II) (Fig. 3c). k$_{cat}$ and K$_M$ could not be calculated for POP$_{167/517}$ in the presence of Ni(II) because saturating substrate concentrations under these conditions were beyond the solubility limit of the substrate. Notably, POP$_{167}$ and POP$_{517}$ displayed only 1.1- and 1.4-fold changes in rate, respectively (Supplementary Fig. 5), demonstrating the necessity of two BpyAla residues for Ni(II)-responsiveness, as expected for inhibition via the intended Ni(II) (BpyAla)$_2$ linkage (Fig. 1a). A 1:1 mixture of POP$_{167}$ and POP$_{517}$ behaved similarly to the individual single-point variants (Supplementary Fig. 5), consistent with the proposed intra-protein cross-link rather than potential inter-protein interactions that

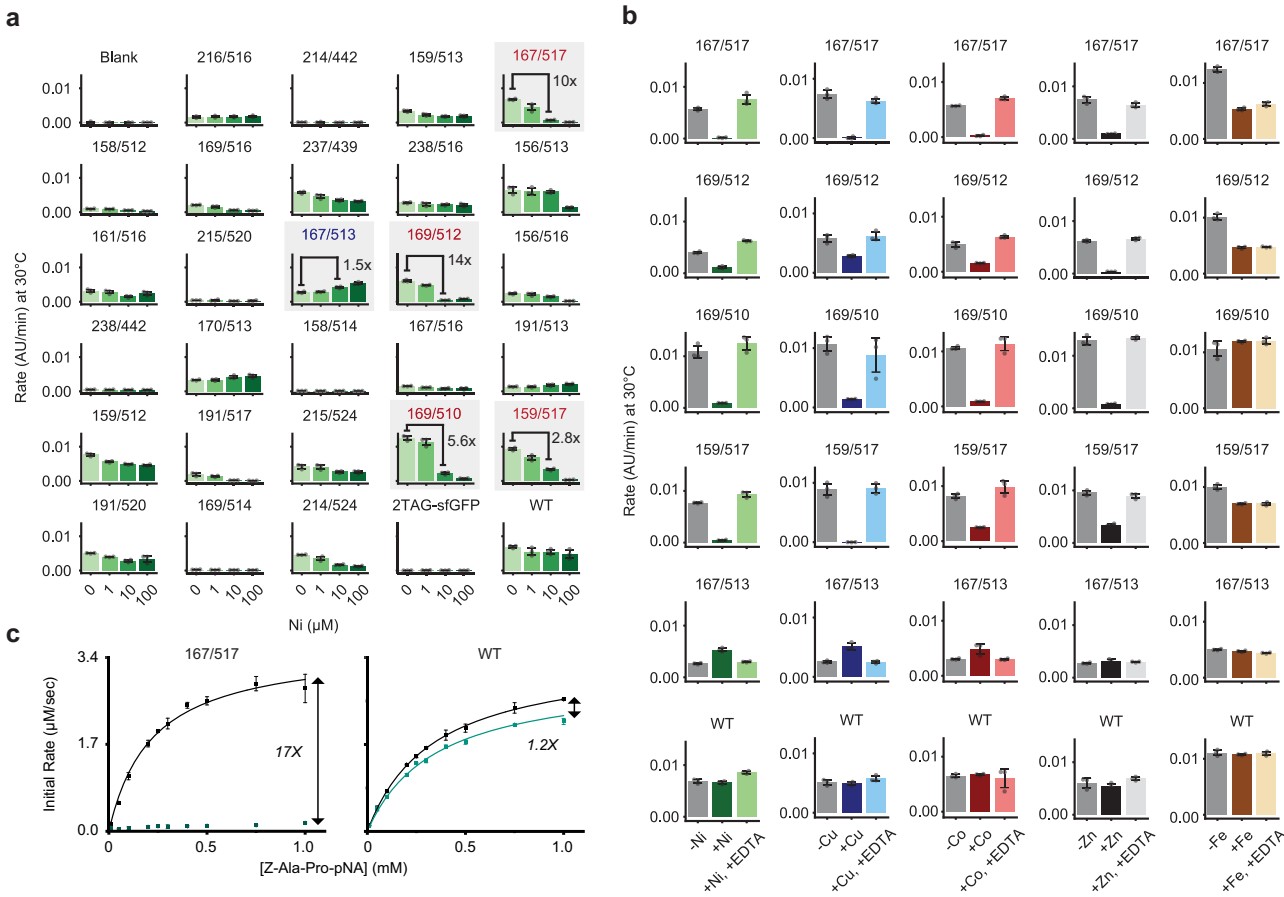

**Fig. 3 Kinetic analysis of POP$_{X/Y}$ variants in response to divalent metals. a, b** The activity of POP$_{X/Y}$ variants from CFPS was assessed on Z-Ala-Pro-$p$NA. **a** Reaction rates (AU/min) at increasing Ni(II) concentrations (0–100 μM). The fold-changes between 0 and 10 μM of Ni(II) for POP$_{167/517}$, POP$_{167/513}$, POP$_{169/512}$, POP$_{169/510}$, and POP$_{159/517}$ are noted. **b** The reversibility of metal-dependent activity upon addition of EDTA shows near-quantitative recovery for Ni(II), Cu(II), Co(II), and Zn(II), while activity is not recovered for changes by Fe(II), Select variants from Fig. 3a are shown. **c** Steady-state kinetic assays performed at 85 °C using purified enzymes (~20 nM) in the presence of either 1 mM EDTA (black) or 5 μM NiCl$_2$ (green). Initial rates (μM/sec) are plotted versus substrate concentration (mM), and data was fit, when appropriate, with the Michaelis–Menten equation. The fold-change between reaction rate at 1 mM substrate is shown. Each data point represents the average of 3 replicates and error bars represent standard deviations. Source data are provided as a Source Data file.

## Table 1 Steady-state kinetic parameters for selected POP and Pluc variants.

| Variant | K$_M$ (μM) | | k$_{cat}$ (sec$^{-1}$) | | V$_{max}$ (RLU) | |
|---|---|---|---|---|---|---|
| | −Ni(II) | +Ni(II) | −Ni(II) | +Ni(II) | −Ni(II) | +Ni(II) |
| [a]POP$_{WT}$ | 335 ± 10 | 345 ± 13 | 172 ± 2 | 152 ± 3 | NA | NA |
| [a]POP$_{167}$ | 326 ± 22 | 399 ± 32 | 136 ± 4 | 131 ± 5 | NA | NA |
| [a]POP$_{517}$ | 196 ± 18 | 197 ± 19 | 154 ± 5 | 110 ± 4 | NA | NA |
| [a]POP$_{167/517}$ | 219 ± 24 | NA | 181 ± 7 | NA | NA | NA |
| [b]Pluc$_{WT}$ | 1.48 ± 0.15 | 1.19 ± 0.15 | NA | NA | 40200 ± 900 | 37900 ± 1000 |
| [b]Pluc$_{202/532}$ | 21.5 ± 1.25 | 2.32 ± 0.44 | NA | NA | 20100 ± 400 | 904 ± 38 |
| [b]Pluc$_{108/508}$ | 0.565 ± 0.145 | 2.19 ± 0.27 | NA | NA | 10600 ± 500 | 1240 ± 40 |

[a]Reactions were conducted in triplicate using 0-1 mM Z-Ala-Pro-$p$NA and 20-21 nM enzyme in 10% v/v DMSO/30 mM HEPES (pH 7.4) containing 0.8 M NaCl at 85 °C for 1 minute. Average initial rates were determined by changes in absorbance over time at 410 nm using a calculated molar extinction coefficient for $p$NA (7,126 M$^{-1}$ cm$^{-1}$). Kinetic parameters were determined by the non-linear regression function in OriginPro using the Michaelis–Menten equation. [b]Reactions were conducted by mixing 1.1 μM enzyme (preincubated with an equal volume of 1 mM Ni (II) when applicable) with 0-0.75 mM D-luciferin in DMSO (5% v/v DMSO total) in 12.5 mM HEPES (pH 7.8), 5 mM MgSO$_4$, and 1 mM ATP. Luminescence was read for 5 min at room temperature immediately after mixing. The maximum RLU values at each D-luciferin concentration, which describe the "glow" phase of the Pluc reaction mechanism[75], were fit to a Michaelis–Menten equation that accounts for inhibition observed at increasing substrate concentrations. This inhibition could be the result of competitive concentrations of inhibitory byproducts[58]. Max rates describe apparent V$_{max}$ of Pluc in the glow phase. Errors for each parameter as calculated in OriginPro from triplicate reactions are provided.

could also result in inhibition. Given the fact that POP$_{167/517}$ provided the highest level of control of any of the variants examined, further characterization of this variant was pursued.

The dynamic range, robustness, and rate of Ni(II)-mediated activity switching for POP$_{167/517}$ were evaluated relative to WT

POP. The enzymes were preincubated with Ni(II) or EDTA before an aliquot was removed to assess activity, which allowed the same enzyme sample to be switched between metallated and apo forms (Fig. 4a). WT POP showed negligible inhibition by Ni(II) and a 1.9-fold decrease in activity between the first and

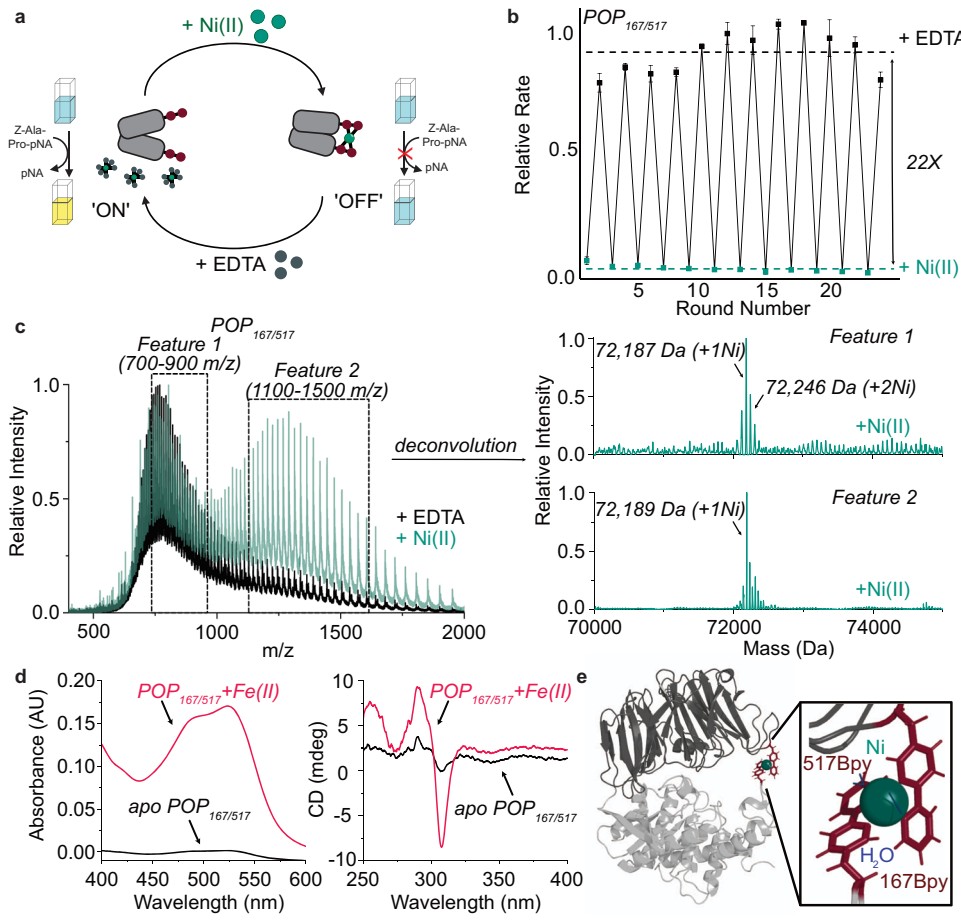

**Fig. 4 Switching and characterization of POP$_{167/517}$. a** Assay for Ni(II)-dependent switching. **b** Ni(II)-dependent switching as measured by the relative rate of Z-Ala-Pro-*p*NA hydrolysis upon addition of Ni(II) (green points) or EDTA (black points); average rates are shown as green or black dashed lines, respectively. Data points are averages of triplicate reactions and error bars represent standard deviations. **c** Intact protein ESI-MS data for POP$_{167/517}$ in the presence of excess EDTA (black) or Ni(II) (green). Raw MS data (left) was deconvoluted (right) using m/z windows of either 700–900 or 1100–1500 for the Ni(II)-treated protein. **d** UV–Vis (left) and CD (right) spectra for apo (black) or Fe(II)-treated (pink) POP$_{167/517}$ showing expected MLCT transition and intraligand charge transfer, respectively, in the presence of Fe(II). **e** A representative structure of Ni(II)-bound POP$_{167/517}$ from MD simulation shows coordination of two BpyAla residues and two waters to the Ni(II) center. Source data are provided as a Source Data file.

final switch, perhaps due to degradation of the enzyme from extended incubation in high concentrations of Ni/EDTA (Supplementary Figure 7). In contrast, POP$_{167/517}$ demonstrated highly effective switching over 24 alternating additions of either Ni(II) or EDTA (Fig. 4b). The enzyme is nearly inactive in the presence of Ni(II), full activity is recovered by the addition of EDTA, and a 22-fold average dynamic range is maintained for the duration of activity cycling. Unlike the WT enzyme, no gradual decrease in the activity towards later cycles was observed, suggesting increased stability. Switching activity on or off was complete in <5 min, highlighting the efficiency of kinetically labile Ni(II)-BpyAla dative bonding for reversibly controlling enzyme activity.

We next aimed to more directly probe Ni(II)-BpyAla binding, since all evidence suggesting the involvement of the intended M(BpyAla)$_2$ linkage had thus far been inferred from the effects of added metal salts on POP$_{X/Y}$ variant activity. Addition of Ni(II) to a solution of apo POP$_{167/517}$ led to a mass spectrum with a low m/z feature consistent with the presence of both one and two Ni(II) ions and a second high m/z feature consistent with the presence of a single Ni(II) ion (Fig. 4c). The higher m/z of the latter feature indicates a lower charge state in which POP$_{167/517}$ residues are less accessible for ionization, as would be expected in a more closed conformation[53], and the single metalation is

consistent with coordination of Ni(II) by both BpyAla residues. Single metalation was also observed for both POP$_{167}$ and POP$_{517}$ in the presence of Ni(II), but only the low m/z feature was observed for these, consistent with metal binding with an open conformation (Supplementary Figures 8 and 9).

Further support for the formation of a M(BpyAla)$_2$ linkage in POP$_{167/517}$ was obtained from circular dichroism spectroscopy and UV–vis spectroscopy (Fig. 4d). While solutions of Ni(II)(Bpy)$_n$ complexes at concentrations compatible with their formation in POP$_{167/517}$ (i.e., <0.5 mM) exhibit minimal absorbance in the UV–vis region[54], the corresponding Fe(II)(Bpy)$_n$ complexes possess a strong MLCT[55] absorption at ~520 nm. This feature was observed upon addition of Fe(II) to solutions of apo POP$_{167}$, POP$_{517}$, and POP$_{167/517}$, and the higher molar extinction coefficient of the latter is consistent with the formation of a Fe(II)(BpyAla)$_2$ linkage (Supplementary Figure 10). Moreover, in the near UV spectral region associated with absorption from aromatic residues like BpyAla, the CD spectrum of POP$_{167/517}$ has a local minimum and maximum at 308 nm and 290 nm (Supplementary Fig. 11), respectively, consistent with the expected intraligand π to π* transition of the ncAA[56]. These Cotton effects are diagnostic of a chiral metal complex[57]. The absence of this feature in POP$_{167}$ or POP$_{517}$ strongly suggests that both BpyAla residues bind Fe(II) in the POP$_{167/517}$ scaffold to

form the intended Fe(II)(BpyAla)$_2$ linkage. The same high m/z feature was observed by intact protein ESI-MS upon addition of Fe(II) to POP$_{167/517}$ as was observed in the presence of Ni(II) (Supplementary Fig. 12), indicating that a similar linkage occurs in the presence of both Ni(II) and Fe(II). Along with data showing inhibition of POP$_{167/517}$ by Fe(II) (Fig. 3b), these data suggest that bidentate metal coordination by two Bpy LGs to generate a M(II)(BpyAla)$_2$ linkage as shown in Fig. 4e favors the closed conformation of POP$_{167/517}$ and prohibits catalytic turnover.

**Extension to luciferase chemical switches.** To demonstrate the generality of the Bpy LG, we used the HTE workflow outlined above to engineer metal-responsive *Photinus pyralis* luciferase (Pluc) variants. Pluc is a genetic reporter used in the life sciences[58], and the ability to switch its activity in response to metal ions may prove useful for biosensing applications. Structurally, Pluc requires a large conformational change between its N- and C-terminal domains to first adenylate luciferin and then catalyze oxygen-dependent decarboxylation to form oxyluciferin and light[59]. MD simulations based on Pluc structures in both catalytic conformations were used to design 22 Pluc variants with BpyAla residues situated at sites that would allow reversible trapping in one conformation or the other (Fig. 5a)[59,60]. Variants were solubly expressed using CFPS with average yields of 600 µg/ mL as determined by $^{14}$C-leucine liquid scintillation counting, and full-length protein was obtained only in the presence of BpyAla, as observed by autoradiography (Supplementary Figure 13).

Metal-dependent regulation of Pluc activity was then examined by incubating the variants with different Ni(II) concentrations and assaying luciferase activity in a saturating amount of luciferin and ATP. Two variants, Pluc$_{202/532}$ and Pluc$_{108/508}$, exhibited good activity (>25% WT activity) and were inhibited in a dose-dependent manner by Ni(II) (Fig. 5b). Steady-state kinetic analysis of these variants in crude CFPS extract indicated that $V_{max}$ and $K_M$ of WT Pluc are only marginally affected by Ni(II),

but $V_{max}$ values for Pluc$_{202/532}$ and Pluc$_{108/508}$ decrease by 20-fold and 8.4-fold, respectively (Table 1). These results are consistent with the proposed conformational trapping preventing the enzyme from catalyzing either adenylation or oxidative decarboxylation. Interestingly, the $K_M$ for luciferin was reduced by a factor of 10 in the presence of Ni(II) for the Pluc$_{202/532}$ variant, which was selected based on the initial luciferin-binding form of the enzyme and could therefore reflect trapping of a conformation better suited to bind luciferin. Total catalytic efficiency ($V_{max}/K_M$) in the presence and absence of metal is 463 and 1005 RLU/µM, respectively, for Pluc$_{202/532}$ and 709 and 19210 RLU/ µM, respectively, for Pluc$_{108/508}$. Our results show that the Bpy LG can be used to regulate the activity of structurally distinct enzymes that catalyze unrelated chemical reactions in a stimulus-driven manner.

## Discussion

In this study, we established an integrated computational and experimental workflow to incorporate a metal-responsive chemical switch into proteins. The workflow uses MD simulations to help minimize experimental design validation and CFPS to synthesize designed variants in high yields sufficient for direct assay. The latter technique enabled a rapid screening approach using HTE to prepare and monitor designs exhibiting different phenotypes (e.g., activation, inactivation, and reversibility) in 384-well plates. While this workflow is limited to enzymes that have substrates, products, or cofactors compatible with high-throughput measurement such as those with optical, fluorescent, or luminescent properties, it nevertheless facilitates screening of many types of enzymes. For example, the activity of any enzyme that oxidizes or reduces NADH could be measured in real-time using this workflow. This HTE approach enabled us to engineer a switch, comprising a pair of genetically encoded BpyAla residues that are strategically incorporated so that reversible metal coordination would bias enzyme conformational states. The Bpy LG was incorporated into dozens of sites in two structurally distinct enzymes, *Pfu* POP and Pluc, with high yields

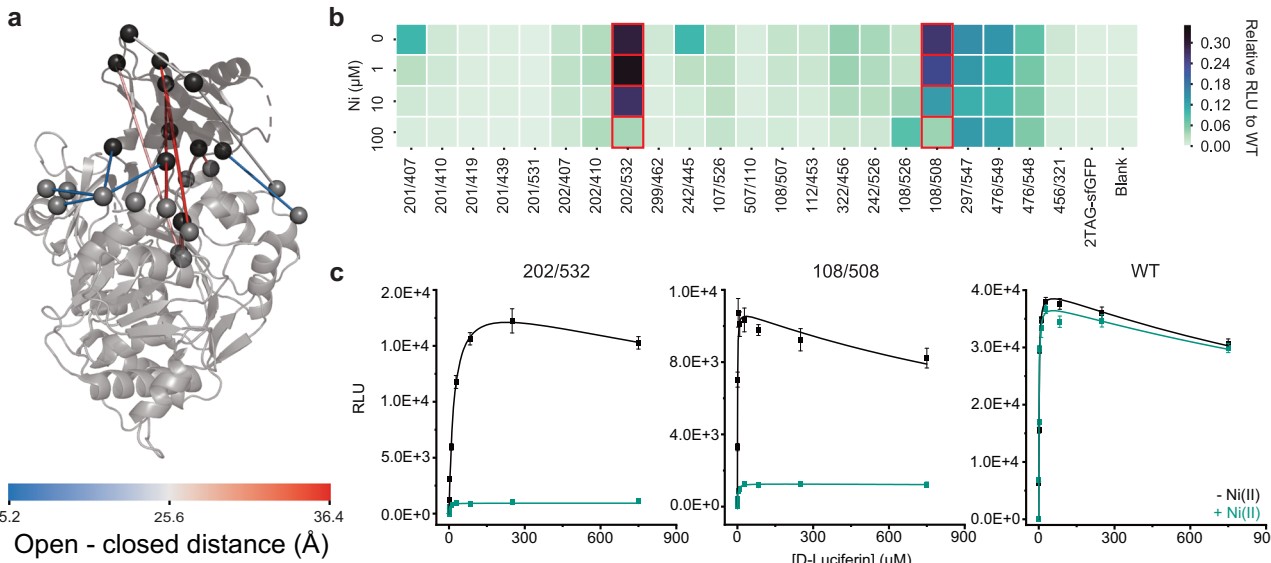

**Fig. 5 Implementation of LG-approach in firefly luciferase. a** Locations of the final 22 LG pairs selected using the procedure outlined in Fig. 2. **b** Pluc variants were synthesized using CFPS and activity of D-luciferin oxidation was assessed. Reaction rates (RLU) are shown in response to increasing Ni(II) concentrations (0–100 µM) as a heat map. Inhibited variants are highlighted in red. **c** Steady-state kinetic assays were performed at room temperature with CFPS-produced enzyme variants Pluc$_{202/532}$, Pluc$_{108/508}$, and Pluc$_{WT}$ in either the absence (black) or presence of Ni(II) (green). Maximum rate (RLU) is plotted versus substrate concentration (µM) and data was fit to a Michaelis-Menten equation that accounts for the observed inhibition using OriginPro. Each data point represents the average of three replicates and error bars represent standard deviations. Source data are provided as a Source Data file.

and only modest impact on the activity of functional variants in the absence of metal, highlighting its functional-orthogonality and potential utility for proteins that may not accommodate large conformationally responsive domains or the introduction of new catalytic sites. The optimal POP variant displays up to a 20-fold dynamic range in response to metal salts and can be switched on/off 24 times with no significant loss in activity, rivaling the best-performing reversible switches reported to date[61]. A similar dynamic range was observed in Pluc, showing the extensibility of LG regulation.

Kinetic, spectroscopic, and computational evidence indicates that switching by Bpy LG pairs involves reversible formation of the intended $M(II)(BpyAla)_2$ complex, which alters enzyme conformational landscapes to inhibit catalytic turnover. In this sense, the genetically encoded BpyAla residue allows facile access to the dynamic metal-ligand bonding with tunable bonding strength and specificity found in naturally occurring metallo-proteins that have far more complex metal coordination environments and regulation transduction networks[62]. BpyAla has been used to generate metallocofactors in artificial metalloproteins[63] and metalloenzymes[64], to stabilize protein motifs like coiled coils[65], and to template protein self-assembly[66], highlighting its propensity to form stable $M(BpyAla)_n$ complexes in a variety of protein contexts. This ability likely underpins the high stability of $POP_{167/517}$ relative to WT POP during activity switching even at elevated temperatures and high salt concentrations and suggests that $M(II)(BpyAla)_2$ complex formation plays a dual structural/functional role similar to metal binding in many natural[67] and some artificial metalloenzymes[68]. Despite its thermodynamic stability, reversible BpyAla-metal binding at domain interfaces distal to enzyme active sites enables metal-responsive[44] allosteric control over catalysis.

Even given the excellent switching properties exhibited by $POP_{167/517}$, several observations suggest that significant improvements to LG switching could be realized. For example, Bpy LG pairs led to the intended metal-dependent inactivation in most POP variants, but some variants were activated (e.g., $POP_{167/513}$). In addition, while the design process outlined in Fig. 2 included geometric constraints from the crystal structures of $Ni(II)(Bpy)_2$ complexes, several variants displayed modest specificity for other metal ions (Supplementary Figure 3). MD simulations also suggest that metal coordination to proximal canonical amino acids like Asn can occur (Supplementary Figure 14). Such interactions, combined with differential Bpy coordination enforced by the POP scaffold, could increase metal affinity or specificity or give rise to unique phenotypes, like the activation observed for $POP_{167/513}$. These findings suggest that metal specificity and regulation behavior could be improved using geometric constraints for different metals, including canonical protein residues in switch designs, and using directed evolution to optimize emergent functions[7–10]. More broadly, while metal-ligand binding by the Bpy LG provides several unique advantages for switching, the methods developed here can be extended to engineer other protein switches involving LGs that respond to different stimuli. These efforts will be enabled by the availability of orthogonal aminoacyl transfer RNA (tRNA) synthetase (aaRS):tRNA pairs suitable for LG incorporation[69,70] and improved methods for incorporating two unique ncAAs[71–74].

Looking forward, we anticipate that genetically encoded chemical switches installed into proteins using the integrated computational/experimental approach described here will enable new strategies for protein engineering and synthetic biology. This capability could expand opportunities for controlling multi-enzyme biocatalysis in vitro[3] for systems that can tolerate added metal salts and chelators. These applications would be facilitated by immobilized chelators or enzymes to minimize accumulation of chelator or both metal and chelator, respectively, since these components could adversely affect particular enzymes. While removal of these components cannot be readily achieved in vivo, it may also be possible to use metal switching for in vivo sensing[14], and other applications[19] where reversible regulation is not required.

## Methods

**High-throughput screening and analysis of POP variants.** CFPS reactions were performed to synthesize all proteins, including WT POP, the entire panel of POP variants, and 2TAG-sfGFP. Only StrepII-tagged enzymes were used for the high-throughput experiments. Blank reactions to dilute reactions to be equimolar were set up in parallel. After reaction completion, CFPS reactions were heat-treated at 75 °C for 15 minutes as a crude purification. Insoluble components were pelleted at 20,000 x g for 10 minutes at 4 °C. The supernatant containing POP enzymes was diluted to 1 μM using blank CFPS reactions as diluent. 4.5 μL of POP enzymes were then mixed with 22.5 μL of 2X POP Buffer (60 mM HEPES pH 7.4, 1.6 M NaCl) 4.5 μL of 0–1 mM metal (Nickel(II) Sulfate Hexahydrate, Copper(II) Sulfate Pentahydrate, Cobalt(II) Chloride Hexahydrate, or Zinc(II) Chloride), and 13.5 μL of water. The mixture was equilibrated on ice for 2 hr. For reversibility screening, 1.8 μL of 50 mM EDTA pH 8.0 was added and then incubated for an additional hour on ice. During the incubations, 1.2 μL of 25 mM Z-Ala-Pro-$p$NA in DMSO was spotted into each well of a clear, flat-bottom 384-well plate from a source plate using the Echo 550. After incubation on ice, the reaction mixture was split into three 13.5 μL aliquots and pre-warmed to 30 °C. Using the Integra Viaflo, 10.8 μL of the reaction mixture was dispensed into the substrate-containing 384-well plate and mixed thoroughly. The plate was quickly spun down and read on a pre-warmed plate reader at 30 °C at 410 nm for two hours.

Kinetic curves were analyzed using a custom Python script. Briefly, this script requires two input files: (i) A minimally formatted raw data CSV and (ii) a descriptor CSV that annotates reaction conditions for each well. The script calculates rate by conducting a linear regression over a sliding window of five timepoints and calculates an average and standard deviation between all replicates. Finally, it identifies and plots the maximum slope for each reaction condition. An example of the script is included in the Supplementary Information.

**Switching assay of POP variants.** The switching of activity between 'on' and 'off' states was performed using a modified version of the kinetic assays described in the Supplementary Information. 500 μL of a 1 μM stock of protein (in MQ H₂O) was incubated at 55 °C with shaking (750 rpm) for the entirety of the assay. For the first round, 1 μL of NiCl₂ solution (10 mM) was added to the protein stock and the sample was incubated for 2 min. An aliquot of protein was removed from the stock and added to a quartz cuvette containing buffer. The amount of protein added to the reaction buffer was adjusted throughout the assay based on the changing concentration due to additions of EDTA/NiCl₂ (Supplementary Table 7). The cuvette was then incubated at 85 °C for 2 min. At the same time, 2 μL of EDTA (10 mM, pH 8.0) was added to the protein stock for the second round and a 2-minute incubation was started. After incubating the cuvette, the reaction was initiated by addition of 100 μL of Z-Ala-Pro-$p$NA (10 mM in DMSO) and the solution was mixed by pipetting up and down several times. The reaction was monitored by following the formation of $p$NA with absorbance measurements (at 410 nm) every 6 s for 1 min. Initial rates were determined by converting absorbance values over time using the molar extinction coefficient ($\varepsilon_{410} = 7{,}126$ M⁻¹ cm⁻¹) in Excel. Simultaneously, an aliquot of protein (20.1 μL) was removed from the protein stock solution and added to a cuvette containing buffer and was incubated at 85 °C for 2 min. This cycle was repeated in the same manner by alternating additions of NiCl₂ and EDTA. Final concentrations of components in the reaction were as follows: 30 mM HEPES (pH 7.4), 100 mM NaCl, 20 nM enzyme, 0.2–30 mM NiCl₂, 0–41 mM EDTA, 1.00 mM Z-Ala-Pro-$p$NA, and 10% (v/v) DMSO. All data were collected in triplicate and averaged. Relative rates were determined by dividing each average initial rate by the maximum average rate collected for that data set so that the highest relative rate was 1. The relative initial rates were plotted versus round to assess the switching of the systems. Error bars represent propagated standard deviations.

**Data collection and analysis.** Protein MS were collected with Waters MassLynx 4.1; UV–vis data for Michaelis–Menten kinetics and POP switching assays were collected with Cary WinUV; UV-Vis spectra for POP variants were collected with Cary WinUV2; CD spectra for POP variants were collected with JASCO Spectra Manager Spectra Measurement; Gen5 Version 2.09.2, Build 2.09.1 was used for plate readers and enzyme kinetics measurements; Typhoon FLA7000 Version 1.2, Build 1.2.1.93 was used for imaging autoradiograms; Echo Plate Reformat Version 1.7.2, Build 24 was used to set up POP enzyme reactions; Microbeta2, Version 1.0 Sp1 was used for scintillation counting for quantification of proteins within CFPS.

Data analysis was conducted using Excel Version 2108, OriginPro 2021 (64-bit) 9.8.0.200 (Academic), PyMOL 2.5.0, Compass Data Analysis 5.1 Build 201.2.4019 (Bruker) for MS data, and Python 3.7 using custom scripts available at DOI: 10.5281/zenodo.6320869.

**Reporting summary**. Further information on research design is available in the Nature Research Reporting Summary linked to this article.

## Data availability
The data used to generate Figs. 2–5 are provided in the Source Data file. Complete experimental methods, characterization, and supplementary data/figures are available in the Supplementary Information. POP structure 5T88 was obtained from the Protein Data Bank. Source data are provided with this paper.

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

## Acknowledgements

This study was supported by the U.S. Army Research Laboratory and the U.S. Army Research Office under Contracts/Grants W911NF-18-1-0034 and W911NF-15-1-0334 (J.C.L.), W911NF-18-1-0181 (M.C.J.), and under Grant Number W911NF-18-1-0200 (J.C.L, M.C.J., and B.R.); the David and Lucile Packard Foundation (M.C.J.); and the Camille Dreyfus Teacher-Scholar Program (J.C.L. and M.C.J.). Y.S.Z gratefully acknowledges receipt of a predoctoral fellowship from the Graduate Training Program in Quantitative and Chemical Biology at Indiana University (T32 GM131994). K.S acknowledges support from a predoctoral fellowship from Chemistry of Life Processes Institute, supported by the National Institute of General Medical Sciences of the National Institutes of Health (T32 GM105538). A.H. acknowledges the Department of Defense National Defense Science and Engineering Graduate Fellowship Program (NDSEG- 36373). We thank Mr. Matthew Jordan for performing ICP-MS measurements; Dr. Jonathan Trinidad for assistance with intact protein ESI-MS; Dr. Giovanni Gonzalez-Gutierrez for assistance with various instrumentation in the IU Physical Biochemistry Instrumentation Facility; Prof. Amar Flood for access to a UV-Vis spectrophotometer; Mr. Saman Shafaie and Dr. Benjamin Owen for training on ESI-MS instruments at IMSERC in NU; and Ashty Karim for helpful discussions on manuscript preparation. The content is solely the responsibility of the authors and does not necessarily represent the official views of the National Institutes of Health.

## Author contributions

Y.S.Z. identified Pluc as a model system for LG switching; developed activity assays for POP and Pluc; designed the POP activity switching experiment; conducted steady state kinetic analysis, activity switching, and characterization of metal binding by POP BpyAla variants; analyzed data; and wrote the manuscript. K.S. developed CFPS protocols to incorporate BpyAla into POP, developed HT assays for POP and Pluc activation, inhibition, and switching; conducted all HT LG design screens; conducted all kinetic analysis of Pluc variant activity; analyzed data; and wrote the manuscript. Y.L. conducted all M.D. simulations and modeling and analyzed data. A.H. assisted in developing HT assays for POP mutant screening. B.L. synthesized BpyAla. B.R. conceived and directed the project, analyzed data, and wrote the manuscript. M.C.J. conceived and directed the project, analyzed data, and wrote the manuscript. J.C.L. conceived and directed the project, analyzed data, and wrote the manuscript.

## Competing interests

M.C.J. is a cofounder of SwiftScale Biologics, Stemloop, Inc., Design Pharmaceuticals, and Pearl Bio. M.C.J.'s interests are reviewed and managed by Northwestern University in accordance with their conflict-of-interest policies. All other authors declare no competing interests.
