## [Peer Review File · Nature Communications]

Reviewers' Comments:

Reviewer #1:

Remarks to the Author:

Key results

The manuscript entitled "Metal-responsive regulation of enzyme catalysis using genetically encoded chemical switches" presents a strategy to control protein function by manipulating naturally occurring conformational changes, in this case domain movements. Using unnatural amino acids and a computationally-guided rational design approach they impressively achieved to control enzyme activity by ~20-fold in a reversible, metal/EDTA-mediated manner. The presented research appears to be original, well planned and highly interesting for the field of protein engineering. Remarkably, the methods section is very meticulously written and should easily enable reproducing the results. In general, I would like to see this manuscript published. However, I have some major concerns regarding the suitability for publication in Nature Communications.

Key concerns

One of my key concerns is that this manuscript does not actually provide what the authors promise to show in the introduction. The significance is set too high. Further flaws can be found in the representation of data, the detail of the state-of-the-art as well as fundamental understanding of biochemical processes. I have provided detailed arguments below sorted into major and minor points. If the editors decide to give the authors a chance for improvement I strongly suggest rewriting major parts as outlined below.

Major points

1) The authors' core selling point for a broad audience is the "key challenge to engineering allosterically regulated enzymes" (Introduction, 2nd paragraph), which they intend to tackle in this publication. I have several major concerns with this:

i) The principle of allostery is not explained at all. It is merely presented as "regulation and control mechanism to ensure that they (enzymes) can respond to environmental stimuli" (Introduction, 1st paragraph). However, "allostery" is an established term to describe that ligand binding to an allosteric site affects ligand binding or enzyme catalysis at the active site. In this regard, engineering true allostery is much more complex than engineering a control mechanism of enzyme activity in response to external stimuli, which seems to be the actual goal of this manuscript.

ii) The authors point out that control mechanisms are "appealing to chemists and synthetic biologists" and that current engineering strategies often "neglect native control capabilities" (Introduction, 1st paragraph). I do not understand this line of argument. For the majority of engineered enzymes it is not important to have a control mechanism. Only when the enzyme is used in a certain application a control mechanism is possibly required. Hence, to work out the main point of interest, this paragraph should focus on WHY control mechanisms are appealing e.g., for which fields of research.

iii) In the 3rd paragraph of the introduction the authors minimalistically provide an overview of how protein function can be rendered sensitive to external stimuli. This paragraph is kept quite superficial and does not reflect the current state-of-the-art in this field. Many advances regarding "functionally-orthogonal methods to control enzyme activity in a stimulus-dependent manner" have been achieved over the last decades. I am especially missing a reference to control mechanisms with light, which are not only highly successful but also include manifold renowned strategies e.g., optogenetics, including allosteric switches or regulation of allostery (see various reviews on light regulation).

iv) In the 4th paragraph of the introduction, the authors explain their objective, which is to use "small molecule switches" incorporated into enzymes through amber codon suppression to control protein conformation and catalysis in a reversible manner. However, they do not demonstrate which achievements have already been made towards this goal. To my knowledge at least the field of light regulation has already achieved some major milestones. The group of Lei Wang has shown to disrupt the conformation and binding capabilities of proteins with a photoswitchable click UAA by controlling the α -helical content (C. Hoppmann, I. Maslennikov, S. Choe, L. Wang, J. Am. Chem. Soc. 2015, 137, 11218. | C. Hoppmann, V. K. Lacey, G. V. Louie, J. Wei, J. P. Noel, L. Wang, Angew. Chem., Int. Ed. 2014, 53, 3932.). The group of Peter Hamm has picked up this concept and controlled ligand binding by a factor of >100-fold with an azobenzene-switch

crosslinked to an α -helix; they could even characterize the signal transmittance (O. Bozovic, B. Jankovic, P. Hamm, Nat. Commun. 2020, 11, 5841. | O. Bozovic, J. Ruf, C. Zanobini, B. Jankovic, D. Buhrke, P. J. M. Johnson, P. Hamm, J. Phys. Chem. Lett. 2021, 12, 4262.). Moreover, the group of Reinhard Sterner light-regulated true allostery in a reversible manner with photoswitchable UAAs at positions distant from the active site by a factor of ~ 10 -fold (A. C. Kneutinger, K. Straub, P. Bittner, N. A. Simeth, A. Bruckmann, F. Busch, C. Rajendran, E. Hupfeld, V. H. Wysocki, D. Horinek et al., Cell Chem. Biol. 2019, 26, 1501–1514.e9. | A. C. Kneutinger, C. Rajendran, N. A. Simeth, A. Bruckmann, B. König, R. Sterner, Biochemistry 2020, 59, 2729.)

v) The authors further state that the key to their approach is “to control protein conformation in ways that mimic native allosteric systems” (Introduction, 4th paragraph). Besides the point that ligand binding is required for allostery, the underlying dynamics of allostery are highly complex and still controversially discussed (e.g. regarding the role of fluctuations versus population shifts). How do the authors attempt to mimic something which is not clearly defined? In my opinion, they should desist from focusing on allostery and instead describe how conformational changes dominate the function of enzymes as well as elaborate their concept by means of a specific example such as POP and luciferase.

vi) Finally, the authors promise that their approach has “no fundamental limits on which proteins and activities could potentially be controlled” (Introduction, 4th paragraph) meaning that it “could be applied to any protein” (Results, chapter 1, 2nd paragraph). This objective is highly ambitious and in order to publish in Nature Communications it should be demonstrated (at least to some degree) that their approach is powerful enough to realize this. To this end, the authors have employed two non-allosteric enzymes, which catalyze different reactions, and which are structurally not related. However, both enzymes exhibit a well-known domain movement, which the authors control by inserting the metal-responsive Bpy-Ala UAA. While I consider this approach highly effective for proteins with such domain motions, the authors did not show that this approach also works for proteins for which such large conformational transitions have not been identified. Since a large percentage of proteins does not show domain movements, I am not convinced that their approach will achieve to control the biological activity of any protein.

Thus, in my opinion the authors’ selling point should be to design a control mechanism of protein activity based on the modulation of protein conformation and not to engineer allostery. In the introduction, they should furthermore clearly lay out the achievements that have already been made towards this goal, especially in the light regulation sector. In this context they should point out the differences to their approach and the potential advantages of metals over light or other external stimuli. In the discussion, they should finally elaborate whether their approach could be used to control any protein and what needs to be improved in order to achieve this.

2) Regarding the significance of the results, I agree with the authors that their approach is highly interesting for the protein engineering field as it shows how proteins can be conformationally manipulated. I am not convinced of the statement that the metal-triggered switch will be useful to control multi-enzyme biocatalysis and in vivo screening (Discussion, last paragraph). First, I consider there could be some problems of high salt concentrations or high EDTA concentrations in multi-step enzyme cascades. Many enzymes are put together in these cascades, which may depend on the presence of metals (hence EDTA might interfere) or which are sensitive to salt. Moreover, an additional purification step would be required to separate the desired product from EDTA, which might not be beneficial for industrial purposes. Second, how do the authors anticipate their approach to work in vivo? EDTA is toxic to cells in μM concentrations (S. Hugenschmidt, F. Planas-Bohne, D. M. Taylor, Archives of Toxicology 1993, 67, 76.), however, the authors use mM concentrations. These statements would have been more convincing when the authors would have shown that their model systems work in a multi-enzyme experiment or in vivo e.g., for firefly luciferase. In a revised manuscript, I suggest that the authors either elaborate in more detail, why they regard the approach suitable for these purposes, or focus on other potential applications.

3) While I enjoyed reading the results section and congratulate the authors for their experimental success, I stumbled upon a few flaws concerning the presentation of the approach, the validity of the data as well as inconsistencies in the data, which in my opinion prohibit immediate publication.

i) Introduction, last paragraph and Results, first chapter.

Despite the clear presentation in Figure 1, there is no point-by-point description of the approach in

the text or the legend to the figure. It is, hence, hard to follow and the approach raises some questions:

- why did the authors use the C_{β} distance in MD simulations
- why were MD simulations required when structures of both open and closed conformations of POP and Luciferase exist
- what is meant by double variants
- how does the metal-triggered switch work and related to this why does the Bpy-Ala side chains need to be located in close distance to each other
- how are the 180,000 potential double variants derived

ii) Results, chapter "HTE screening of POP switches", 1st paragraph

The authors state that all POP variants were produced at full-length and without truncation products (legend to Supplementary Figure 2). However, supplementary Figure 2 does not prove this statement. In the autoradiogram in b) no protein marker was used to show that the molecular weight range is covered. In fact, some weak bands are visible for the majority of variants, which are missing for the wt POP. In addition, the MS data do also not cover the molecular weight range of potential truncation fragments e.g., POP₂₁₆ ~ 25 kDa or POP₅₁₆ ~ 59 kDa. Showing the ratio of full-length versus truncated protein is crucial when the protein was not further purified.

iii) Results, chapter "HTE screening of POP switches", 1st paragraph

A high-throughput screening system was set up to "enable screening and characterization of hundreds of enzyme variants and reactions in less than a day." However, the authors have produced 27 and not hundreds of variants to test. Moreover, with regard to the ambition to apply the approach to any protein, it will be difficult or hardly possibly for many other enzymes to set up such a HTE screening since not all reactions allow a spectrophotometric detection. It should be discussed in the discussion section how this step can be adapted for other proteins e.g., for those without the possibility to monitor activity spectrophotometrically.

iv) Results, chapters "HTE screening of POP switches" and "Functional characterization of a BbyAla POP switch", throughout

Of the 27 tested variants, the authors have identified three, POP_{167/517}, POP_{169/510}, and POP_{159/517}, with Ni(ii)-mediated inhibition of POP activity (2nd paragraph, line7). Unfortunately, the results are not presented consistently for these three variants:

- Figure 3a only highlights POP_{167/517}, and POP_{159/517} but not POP_{169/510}
- the decrease factor in line 10 is also only given for POP_{167/517}, and POP_{159/517}
- SDS-PAGE gels for purified POP is only shown for POP_{167/517}, although steady-state kinetics of purified POP_{159/517} (Figure 3c; Table 1) and POP_{169/510} (Supplementary Figure 6) are shown as well
- the steady-state kinetics of POP_{169/510} is only shown in Supplementary Figure 6 without any mentioning thereof in the main text
- in Supplementary Figure 6 a fourth variant appears, POP_{169/512}, which is also not mentioned in the main text
- two controls POP₁₆₇ and POP₅₁₇ for POP_{167/517} were analyzed in steady-state kinetics, however, appropriate control variants for the other two variants POP_{159/517} and POP_{169/510} are missing
- further characterizations after steady-state kinetics were only performed for POP_{167/517}

The authors should refrain from jumping between four, three, two or only one variant by supplementing the missing data and clearly explaining why for the full characterization only POP_{167/517} is regarded (I suspect because it showed the highest control factor).

v) Results, chapter "Functional characterization of a BpyAla POP switch", 1st paragraph

In steady-state kinetics of POP_{167/517} the authors do not give the K_M and k_{cat} in presence of Ni(II) without further statement. The graph in Figure 3c is quite small so it is not apparent for the reader why the constants could not be determined. I suspect the K_M value is too high to be resolved. However, this should be made clear and also, if this is the case, why they did not increase the substrate concentration range to determine the constants. If increasing substrate concentration has no practical limitations, I strongly suggest to re-measure the steady-state kinetics.

vi) Results, chapter "Functional characterization of a BpyAla POP switch", last paragraph
I understand why the authors used iron to visualize the metal-BpyAla linkage in the UV/Vis. However, to correctly conclude that the formation of the linkage is also the cause for a drop in activity, the authors must show that inhibition of POP also works with Fe(II). They have shown successfully that inhibition works with Ni(II), Cu(II), Co(II) and Zn(II) in Supplementary Figure 4 but Fe(II) data are missing. The data should be provided before publication.

vii) SI, chapter "Physical Characterization of POP Variants – Circular Dichroism", p.45

The integrity of the secondary structure of POP variants in absence and presence of Fe(II) cannot be properly validated. The representation should focus on the far UV range of ~185 to ~260 nm. From what is shown, two issues should be addressed:

- The noise below 200 nm seems to be significant and hence no clear transition at ~190 nm indicative of the $\pi \rightarrow \pi^*$ transition of peptide bonds is visible. Since the protein is stored in H₂O no buffer components cause this effect. The authors should check the High Tension voltage traces. If these indicate overloading, the CD spectra should be remeasured with lower protein concentration or in a thinner cuvette to prove the structural integrity.
- The signal strength at ~220 nm differs significantly between the samples, which might be indicative of a decrease in secondary structure potentially caused by protein degradation. It might also point to variations in protein concentration. The authors should check this.

Minor points

Introduction, last paragraph: "...used MD simulations to identify all residue pairs that undergo significant changes... during catalysis."

- Rephrase. To my knowledge the actual catalysis event cannot be simulated with MDs.

Introduction, last paragraph: "...to select functional candidates for analysis."

- Clarify what is meant by "functional candidate".

Results, chapter "Model selection and design of POP switches", 1st paragraph

- The authors talk once of "domains" and then again of "subunits". Clearly the β -propeller and the peptidase parts of POP are domains since they are located on the same protein chain. The term subunits is used for two separate protein chains. Replace "subunits" with "domains".

Results, chapter "Model selection and design of POP switches", Figure 2

- which PDB was used for the representation of POP?

- in a) "Placement between domains" has not been mentioned as a criterium in the main text

- in b) Is the Zn(Bby)₂ Ligand correctly oriented? Further guidance including atom-wise coloring and orientation similar to the Bby-Ala in Figure 1a would help to understand how the 10.4 Å distance was derived.

Results, chapter "Model selection and design of POP switches", 2nd paragraph

"(i) ... in one enzyme conformational state," - in which state? Closed or open?

"(iii) undergo suitable changes in C β distance to prohibit bond formation a different conformational state." - Clarify this point. I understand this criterium that metal binding should only occur in one conformation but not the other and that hence the two Bby-Ala positions need to be close in one conformation but distant in the other.

Results, chapter "HTE screening of POP switches", 1st paragraph, "... at an average yield of 1098 +/- 124 μ g/mL..."

- Define μ g/mL. Do the authors mean the yielded concentration or do they mean 124 μ g protein yielded from 1 mL expression volume?

Results, chapter "HTE screening of POP switches", 2nd paragraph, "... understanding allosteric effects on protein function."

- Replace "allosteric" with "conformational".

Results, Supplementary Figure 3&4

- Highlight the three variants POP_{167/517}, POP_{169/510} and POP_{159/517} as in Figure 3a

Results, chapter "HTE screening of POP switches", 3rd paragraph, "... showing the weakest degree

of inhibition or activation..."

- Give quantitative numbers as in Figure 3a in the main text and Supplementary Figure 3.

Results, chapter "Functional characterization of a BbyAla POP switch", Supplementary Figure 5

- Full-length and truncated protein should be highlighted in the figure or the molecular weight given in the legend.

Results, chapter "Functional characterization of a BbyAla POP switch", Figure 3c

- What is meant by the labels "167N517V" and "159K517V"?

Results, chapter "Functional characterization of a BbyAla POP switch", Table 1

- Provide the standard error of the Michaelis-Menten fit as calculated by Origin.

Results, chapter "Extension to luciferase chemical switches", Figure 5

- Why does luciferase show a substrate inhibition effect? Is this known for the enzyme? Please clarify in the main text.

SI, chapter "MD simulations of POP" and "MD simulations of Luciferase" p. 26

- For reproducibility reasons please provide the PDB-IDs for POP and luciferase structures used in MD simulations

- Clarify in more detail how the BpyAla side chains were introduced.

- What is meant by "cross-linked luciferase"? Please explain.

SI, chapter "Cell extract preparation" p. 31

- pEVOL vectors contain constitutive promoters for aaRS1 and tRNA as well as an araC induction system for aaRS2. However, here IPTG was used for induction. Does pEVOL-BpyRS differ from other pEVOLs?

SI, chapter "Steady-State Kinetic Assays" p.39-41

- "For steady-state kinetic assays with variants POP167/513, POP169/510, POP169/512, and POP167/516, protein samples ..." Three of the variants have never been mentioned in the main text, however, POP159/517 is missing.

- Why were only POP wt, POP167, POP517, and POP167/517 purified after EDTA treatment by SEC? Could diafiltration for the other variants leave traces of EDTA, which might have inhibited POP activity in the kinetic assays?

- Why was POP transferred to pure H₂O. Most proteins require a buffer system for stability, why not POP?

- Why does Supplementary Figure 19 show exemplary kinetic traces for POP167/513 which is never mentioned in the main text?

- The data for POP167, POP517 and POP169/510 in Supplementary Table 6 should be transferred to Table 1 or at least referred to in the main text.

- Data for POP169/512 and POP167/513 in Supplementary Table 6 are irrelevant because they have never been mentioned in the main text.

Reviewer #2:

Comments:

Zubi, et al demonstrated that the formation of $M(\text{bpy})_2$ species can endow a genetically encoded chemical switch to regulate the catalytic activities. They conducted structural analysis and MD simulation to assess the desirable positions to incorporate a pair of bpy-Ala residues. Consequently, 24 POP variants were prepared by cell-free protein expression systems, and at least 12 ones are responsive to the first-row transition divalent metal ions, such as Co, Ni, Cu, and Zn. In addition, the proof of concept was applied to luciferase, expanding the scope of systems.

1) One of the general prerequisites for any chemical switch would be not to perturb the native catalytic activity and protein stability. Otherwise, it would be no longer useful. In this regard, the catalytic activities of Pluc variants were substantially perturbed from the wild-type, yielding 25% or 50% of the V_{max} even in the absence of Ni ion. Can authors speculate how such a significant loss of activities results from the incorporation of bpy-Ala residue? Perhaps, are bpy-Ala residues incorporated in the vicinity to the catalytically essential residues? Based on these results, can authors improve the designing scheme to propose the metal-dependent switches?

2) The authors carried out structural analysis and MD simulation to design 24 POP variants to create a pair of bpy-Ala variants. Although all satisfied the geometric constraints in the designing process, their response to metal ions varies significantly. To provide a better guideline, additional biochemical characterization would be necessary to speculate why some variants worked but not others. Perhaps, the authors can group 24 POP variants into four categories, one showing metal-dependent activity as expected, inverse metal-dependent activity, no metal-dependence with catalytic activity, and no metal-dependence/activity. Then, the authors may characterize any representative variants from each group to explain their distinct behavior on metal ions.

3) 167/513 and 170/513 show inverse metal-dependent activity in that they exhibit even accelerated catalytic activity upon the addition of metal ions. How is it even possible?

4) To my best knowledge, POP is a dimeric protein. Therefore, the addition of bpy-Ala residue at one position of a protomer would form not only $M(\text{bpy})_2$ moiety but also two distantly located bpy residues. Although it would influence both structure and function of the proteins, the authors neither described nor discussed this aspect in this manuscript. Because these residues are exposed to the protein surface, the latter bpy-Ala may form intermolecular complexes, possibly generating undesirable high-ordered species. Is it why the SDS-PAGE analysis was carried out in the presence of excess EDTA? Perhaps, the formation of high-ordered species might be related to their distinct response to the metal ion. Do authors examine the oligomeric structures of the proteins upon the addition of metal ions? Perhaps, size exclusion chromatography or analytical ultracentrifugation in the presence of Ni can be carried out to quantify the fraction of catalytically active and metal-responsive species.

5) If the intermolecular complexation occurs, as shown below, what are the steady-state kinetic

parameters of the 1:1 mixture of the 167 and 517 variants?

6) The authors characterized one of the POP variants by UV-Vis and CD spectroscopy, demonstrating that Fe would be a surrogate metal element to show Ni-dependent chemical switch. However, neither ferric nor ferrous ion was not applied for this system in Figure 3 for no apparent reason. Please provide the catalytic activities of POP variants with Fe ion.

7) Please include why the authors used a cell-free expression system instead of standard e coli heterologous expression. Is it to increase protein yields of having two bpy-Ala residues in a protomer? More detailed information in the experimental setup might be necessary for readers.

8) Among the divalent metal ions that the authors used, the cupric ion is redox-active under physiological conditions, and reduction may lead to significant changes in the geometry of $M(\text{bpy})_2$ and subsequent metal-dependent catalytic activities. Because the catalytic activities of POP are independent of dioxygen, the function of the metal-dependent switch can be monitored under anaerobic conditions. Can authors explore whether these systems are redox-dependent?

9) Do authors only consider the distance between two residues for MD simulation? What about the directionality of the side-chains? Now that the authors have both simulation and experimental data sets, can any conclusion or criteria in the determination of two bpy-positions can be drawn and reassessed?

10) Figure 2C has no information related to residues. Please provide the residue labels. In addition, add the distance between i and j residues in Supplementary Table 2. Please also include a column showing whether the pairs are responsive to metal ions as expected or not.

11) The metal-dependent chemical switch can be valuable, but it would not be applicable for every enzyme system. The limitation in the scope of the enzyme should be discussed. For example, metalloenzymes, having a metal cofactor are unlikely to be applicable for this chemical switch because it can be removed by EDTA treatment. In addition, only the enzymes having sufficient conformational changes at the region of substrate access and product release step can be applied.

12) There is no standard deviation or experimental error in Table 1 and Supplementary Table 6. Please provide the values.

13) Although $M(\text{bpy})_2$ binding constants are fairly high, the authors always used metal ions in great excess. Is there any reason for using excess metal elements? For reversibility experiments, it would be better to use the minimum amounts of the metal element. Did the authors observe any variations in the binding affinity depending on the variants?

Point-by-point Response to Reviewers – Zubi et al.

Reviewer #1 (Remarks to the Author):

Key results

The manuscript entitled “Metal-responsive regulation of enzyme catalysis using genetically encoded chemical switches” presents a strategy to control protein function by manipulating naturally occurring conformational changes, in this case domain movements. Using unnatural amino acids and a computationally-guided rational design approach they impressively achieved to control enzyme activity by ~20-fold in a reversible, metal/EDTA-mediated manner. The presented research appears to be original, well planned and highly interesting for the field of protein engineering. Remarkably, the methods section is very meticulously written and should easily enable reproducing the results. In general, I would like to see this manuscript published. However, I have some major concerns regarding the suitability for publication in Nature Communications.

We appreciate the reviewer's support of publication in Nature Communications following suitable revision.

Key concerns

One of my key concerns is that this manuscript does not actually provide what the authors promise to show in the introduction. The significance is set too high. Further flaws can be found in the representation of data, the detail of the state-of-the-art as well as fundamental understanding of biochemical processes. I have provided detailed arguments below sorted into major and minor points. If the editors decide to give the authors a chance for improvement I strongly suggest rewriting major parts as outlined below.

We thank the reviewer for raising these key concerns. In the text that follows and per the reviewer's suggestions, we have revised major parts of text to avoid overstating significance and to clarify data representation.

Major points

1) *The authors' core selling point for a broad audience is the “key challenge to engineering allosterically regulated enzymes” (Introduction, 2nd paragraph), which they intend to tackle in this publication. I have several major concerns with this:*

i) The principle of allostery is not explained at all. It is merely presented as “regulation and control mechanism to ensure that they (enzymes) can respond to environmental stimuli” (Introduction, 1st paragraph). However, “allostery” is an established term to describe that ligand binding to an allosteric site affects ligand binding or enzyme catalysis at the active site. In this regard, engineering true allostery is much more complex than engineering a control mechanism of enzyme activity in response to external stimuli, which seems to be the actual goal of this manuscript.

ii) The authors point out that control mechanisms are “appealing to chemists and synthetic biologists” and that current engineering strategies often “neglect native control capabilities” (Introduction, 1st paragraph). I do not understand this line of argument. For the majority of engineered enzymes it is not important to have a control mechanism. Only when the enzyme is used in a certain application a control mechanism is possibly required. Hence, to work out the main point of interest, this paragraph should focus on WHY control mechanisms are appealing e.g., for which fields of research.

We appreciate reviewer for raising this point of clarification. We reframed the introduction so that we avoid using the term allostery to describe our system, and instead focus on controlling conformation of proteins. We also added several examples where control mechanisms have

Point-by-point Response to Reviewers – Zubi et al.

been shown to be/could be important for biocatalysis/chemical synthesis. Both issues raised primarily concern paragraph 1. This has been rewritten as follows:

Naturally occurring enzymes have evolved to catalyze chemical reactions, in many cases, with exquisite selectivity, substrate specificity, and high catalytic rates.¹ Many enzymes also possess regulation and control mechanisms to ensure that they can respond to environmental stimuli.² Leveraging these remarkable properties for chemical synthesis has long appealed to chemists and synthetic biologists,^{3,4} but current enzyme engineering strategies largely focus on improving catalytic properties or changing substrate specificity.⁵ These engineering strategies often neglect native control capabilities like allosteric regulation, which can be inefficient for non-native substrates,⁶ or lost without selection pressure as has been observed in several directed evolution approaches^{7–10}. Most engineered enzymes are used in highly optimized single-reaction processes^{11,12} where allosteric control is not needed and could even be detrimental. The need for modular, orthogonal methods to control enzyme activity in a stimulus-dependent manner has become increasingly apparent,¹³ however, for efforts to evolve proteins for in vitro multi-enzyme cascade catalysis¹⁴, diagnostic tools,^{15–17} biosensing,^{18,19} and construction of novel signaling circuits²⁰

iii) In the 3rd paragraph of the introduction the authors minimalistically provide an overview of how protein function can be rendered sensitive to external stimuli. This paragraph is kept quite superficial and does not reflect the current state-of-the-art in this field. Many advances regarding “functionally-orthogonal methods to control enzyme activity in a stimulus-dependent manner” have been achieved over the last decades. I am especially missing a reference to control mechanisms with light, which are not only highly successful but also include manifold renowned strategies e.g., optogenetics, including allosteric switches or regulation of allostery (see various reviews on light regulation).

We agree that we should have better highlighted past studies focused on the development of functionally-orthogonal methods to control enzyme activity in a stimulus-dependent manner. Per the reviewer’s suggestion, we have now added more detail to describe methods used to control protein dynamics and conformations with light and cite two recent reviews. We felt that the light dependent systems were best described in paragraph 2, which has been rewritten as follows:

A key challenge to engineering dynamically regulated enzymes is the complex interplay between catalytic activity, substrate specificity, and regulation. For example, while native allosteric proteins have been engineered to respond to new stimuli, this control typically applies to the native protein function, frequently DNA binding.^{6,21} Chimeric systems have also been developed to couple a naturally responsive protein to a protein of interest (POI) so that regulation of the former allows for control over the latter²². Optical regulation is particularly notable in this regard given that it allows for precise spatiotemporal control of protein function²³. This approach is commonly achieved by fusing a POI to the light, oxygen, and voltage (LOV) responsive domain of plant phototropins, which can control protein function through blue-light induced conformational changes (e.g. sterically hindering accessibility to enzyme active sites or control ordered and disordered protein regions).²⁴ Optogenetic dimerization has been successfully implemented in systems that utilize LOV domains^{25,26} and those based on Arabidopsis thaliana cryptochrome 2 (CRY2)²⁷. These approaches often require that large (potentially disruptive) regulatory domains be used,^{28,29} and achieving efficient regulatory

Point-by-point Response to Reviewers – Zubi et al.

transduction between domains can require extensive protein engineering³⁰. Finally, de novo design has been used to produce switchable proteins,^{31,32} but switching has only been reported in response to peptides and proteins, and de novo design of switchable enzymes has not been reported.

iv) In the 4th paragraph of the introduction, the authors explain their objective, which is to use “small molecule switches” incorporated into enzymes through amber codon suppression to control protein conformation and catalysis in a reversible manner. However, they do not demonstrate which achievements have already been made towards this goal. To my knowledge at least the field of light regulation has already achieved some major milestones. The group of Lei Wang has shown to disrupt the conformation and binding capabilities of proteins with a photoswitchable click UAA by controlling the α -helical content (C. Hoppmann, I. Maslennikov, S. Choe, L. Wang, J. Am. Chem. Soc. 2015, 137, 11218. | C. Hoppmann, V. K. Lacey, G. V. Louie, J. Wei, J. P. Noel, L. Wang, Angew. Chem., Int. Ed. 2014, 53, 3932.). The group of Peter Hamm has picked up this concept and controlled ligand binding by a factor of >100-fold with an azobenzene-switch crosslinked to an α -helix; they could even characterize the signal transmittance (O. Bozovic, B. Jankovic, P. Hamm, Nat. Commun. 2020, 11, 5841. | O. Bozovic, J. Ruf, C. Zanobini, B. Jankovic, D. Buhrke, P. J. M. Johnson, P. Hamm, J. Phys. Chem. Lett. 2021, 12, 4262.). Moreover, the group of Reinhard Sterner light-regulated true allostery in a reversible manner with photoswitchable UAAs at positions distant from the active site by a factor of ~10-fold (A. C. Kneuttinger, K. Straub, P. Bittner, N. A. Simeth, A. Bruckmann, F. Busch, C. Rajendran, E. Hupfeld, V. H. Wysocki, D. Horinek et al., Cell Chem. Biol. 2019, 26, 1501–1514.e9. | A. C. Kneuttinger, C. Rajendran, N. A. Simeth, A. Bruckmann, B. König, R. Sterner, Biochemistry 2020, 59, 2729.)

To address the reviewers concern, we have now added several references with respect to innovations and milestones from modulating protein activity with light. We have added the references provided by the reviewer:

- C. Hoppmann et. al. J. Am. Chem. Soc. 2015, 137, 11218.
- A. C. Kneuttinger et. al. Cell Chem. Biol. 2019, 26, 1501–1514.e9.; A. C. Kneuttinger et al Biochemistry 2020, 59, 2729.

We have decided to not include the work by Peter Hamm as these papers do not use ncAAs to accomplish their work but use cysteines to ligate on an azobenzene switch into the proteins of interest. We also added important context to clarify differences between these earlier studies and our results. These edits were in paragraph 3, which has been rewritten as follows:

*We envisioned that small molecule switches that undergo reversible formation or cleavage of covalent bonds in the presence of different stimuli could be integrated into enzymes to enable regulation of catalysis. For yexample, boronic acids and diols form boronic esters at low pH,³² hydrazines and aldehydes form hydrazones in the presence of anilines,³³ and bipyridines bind metal ions to form bis- or tris-bipyridine complexes³⁴. We reasoned that amber codon suppression methods³⁵ could be used to genetically encode non-canonical amino acids (ncAAs) that contain the reactive functional groups (linking groups, LGs) found in synthetic switches. *Previous studies have established that ncAAs can be integrated into proteins and peptides to reversibly control conformation.*³⁶ Recently it has also been shown that ncAAs can be used to control protein function,^{37,38} although most examples rely on irreversible mechanisms like deprotection of caged*

Point-by-point Response to Reviewers – Zubi et al.

residues^{39,40,42}. Reversible control of protein function has been demonstrated by incorporating light-responsive ncAAs, such as those based on azobenzenes, into an enzyme that is already subject to allosteric regulation.^{41,42} Key to our approach, however, is the use of LGs to control the conformation of proteins that are not already subject to allosteric control to enable reversible control over enzyme catalysis. LG incorporation would involve minimal disruption of protein structure and would only require that the LGs can be placed in an orientation to disrupt catalysis upon metal binding. The structural simplicity of the LG approach, however, belies the potential difficulty of engineering systems in which two LGs impart suitable switching properties.

v) The authors further state that the key to their approach is “to control protein conformation in ways that mimic native allosteric systems” (Introduction, 4th paragraph). Besides the point that ligand binding is required for allostery, the underlying dynamics of allostery are highly complex and still controversially discussed (e.g. regarding the role of fluctuations versus population shifts). How do the authors attempt to mimic something which is not clearly defined? In my opinion, they should desist from focusing on allostery and instead describe how conformational changes dominate the function of enzymes as well as elaborate their concept by means of a specific example such as POP and luciferase.

Metal ions are ligands that bind to Bpy LGs. Ligand binding occurs outside of the active site and regulates enzyme activity, so our system meets the basic criteria for allostery—specifically metalloallostery. That being said, our intent was to focus on the control that the resulting allostery provided as the reviewer suggested. To this end, we have taken the reviewer’s advice regarding the entire introduction as outlined above and in the passage noted in comment (v) to focus on controlling activity rather than any specific mechanism. We instead bring this idea up only in the discussion and as more of an analogy. In response to comment (v), we clarified (in paragraph 3, see above) how our approach allows for reversible control of enzyme catalysis (unlike the previously cited examples that allow for irreversible activation) in enzymes that were not natively subject to allosteric control (unlike the new cited systems from Kneuttinger et. al.).

vi) Finally, the authors promise that their approach has “no fundamental limits on which proteins and activities could potentially be controlled” (Introduction, 4th paragraph) meaning that it “could be applied to any protein” (Results, chapter 1, 2nd paragraph). This objective is highly ambitious and in order to publish in *Nature Communications* it should be demonstrated (at least to some degree) that their approach is powerful enough to realize this. To this end, the authors have employed two non-allosteric enzymes, which catalyze different reactions, and which are structurally not related. However, both enzymes exhibit a well-known domain movement, which the authors control by inserting the metal-responsive Bpy-Ala UAA. While I consider this approach highly effective for proteins with such domain motions, the authors did not show that this approach also works for proteins for which such large conformational transitions have not been identified. Since a large percentage of proteins does not show domain movements, I am not convinced that their approach will achieve to control the biological activity of any protein.

Thus, in my opinion the authors’ selling point should be to design a control mechanism of protein activity based on the modulation of protein conformation and not to engineer allostery. In the introduction, they should furthermore clearly lay out the achievements that have already been made towards this goal, especially in the light regulation sector. In this context they should point out the differences to their approach and the potential advantages of metals over light or

Point-by-point Response to Reviewers – Zubi et al.

other external stimuli. In the discussion, they should finally elaborate whether their approach could be used to control any protein and what needs to be improved in order to achieve this.

As outlined above, we have taken steps to mitigate much of these concerns, switching our focus to designing a modular control mechanism, as suggested by the reviewer. We now note how metal regulation could be useful in the final paragraph of the introduction and specify that our initial system relies on enzymes that possess catalytically relevant conformational dynamics in both the introduction and the second paragraph of the results section. Finally, we included additional discussion on how we envision expanding this method to regulate enzymes that do not undergo large conformational movements like those systems selected for our initial studies.

Paragraph 4 from the introduction:

To address this difficulty, here we developed a framework to design, build, test, and analyze enzymes with genetically encoded chemical switches. Initial efforts focused on bipyridine (Bpy) LGs that could induce reversible activation of enzymes in the presence/absence of metal salts (Figure 1a). Specifically, bis-bidentate metal binding by the Bpy LGs would restrict the enzyme to a closed conformation incapable of turnover while removal of the metal would allow the enzyme to access conformational changes required for turnover. We envisioned that this mode of regulation could be used in vitro to cycle enzyme activity or under a broader range of situations where a single metal-dependent activation or deactivation is needed. Our framework (Figure 1b) used molecular dynamics (MD) simulations to identify all residue pairs that undergo significant changes in C_{β} distance between POP conformations. Enzyme variants containing LG pairs at candidate sites were then designed and rapidly prototyped using cell-free protein synthesis (CFPS)⁴³ and high-throughput experimentation (HTE) to select active enzyme variants with metal-dependent activity for analysis. Metal-responsive variants⁴⁴ of two distinct enzymes not subject to native allosteric control were thus engineered by disrupting catalytically relevant conformational dynamics via LG-metal binding. This approach complements existing methods for tuning enzyme catalysis and providing a means to genetically encode functionally orthogonal control elements to regulate enzyme catalysis.

Paragraph 2 from the results section showing initial focus on enzymes that undergo conformation change:

POP is a 72 kDa protein with 189,420 possible double variants. As many proteins of interest could have a similarly large number of potential LG site pairs, or more, we developed a computationally guided approach for identifying sites for LG incorporation that could be applied to any protein that undergoes catalytically relevant conformational changes.

2) Regarding the significance of the results, I agree with the authors that their approach is highly interesting for the protein engineering field as it shows how proteins can be conformationally manipulated. I am not convinced of the statement that the metal-triggered switch will be useful to control multi-enzyme biocatalysis and in vivo screening (Discussion, last paragraph). First, I consider there could be some problems of high salt concentrations or high EDTA concentrations in multi-step enzyme cascades. Many enzymes are put together in these cascades, which may depend on the presence of metals (hence EDTA might interfere) or which are sensitive to salt. Moreover, an additional purification step would be required to separate the desired product from

Point-by-point Response to Reviewers – Zubi et al.

EDTA, which might not be beneficial for industrial purposes. Second, how do the authors anticipate their approach to work *in vivo*? EDTA is toxic to cells in μM concentrations (S. Hugenschmidt, F. Planas-Bohne, D. M. Taylor, *Archives of Toxicology* 1993, 67, 76.), however, the authors use mM concentrations. These statements would have been more convincing when the authors would have shown that their model systems work in a multi-enzyme experiment or *in vivo* e.g., for firefly luciferase. In a revised manuscript, I suggest that the authors either elaborate in more detail, why they regard the approach suitable for these purposes, or focus on other potential applications.

We thank the reviewer for highlighting that our approach is highly interesting. The utility of any reversible switch for an *in vitro* cascade reaction will depend on the specific enzymes used and the specific reaction conditions associated with those enzymes. The goal of our study is to report the development of an approach that users can evaluate relative to other systems. It may be that for some enzymes, the use of metals and chelating agents could be problematic, but there are also engineering approaches to address these issues (e.g. metal chelating resins that can be removed, immobilized enzymes, etc.). In our opinion, that level of engineering is beyond the scope of this report, but we hope to move beyond this to pursue such applications in due course. We agree with the reviewer that *in vivo* switching would be difficult for the reasons noted and changed the text to focus on *in vitro* cascades as outlined above. The *in vivo* applications we had in mind involved inhibition of enzyme activity that could be removed *in vitro*. For example, expression of toxic proteins that are inactivated by metals *in vivo* (since our proteins express in metal-bound form) could then be used in the absence of metal *in vitro*. Likewise, sensing applications would not require reversible activation. We have modified the final paragraph to make this clear as shown here:

*Looking forward, we anticipate that genetically encoded chemical switches installed into proteins using the integrated computational/experimental approach described here will enable new strategies for protein engineering and synthetic biology. This capability could expand opportunities for controlling multi-enzyme biocatalysis *in vitro*³ for systems that can tolerate added metal salts and chelators. These applications would be facilitated by immobilized chelators or enzymes to minimize accumulation of chelator or both metal and chelator, respectively, since these components could adversely affect particular enzymes. While removal of these components cannot be readily achieved *in vivo*, it may also be possible to use metal switching for *in vivo* sensing¹⁵, and other applications²⁰ where reversible regulation is not required.*

3) While I enjoyed reading the results section and congratulate the authors for their experimental success, I stumbled upon a few flaws concerning the presentation of the approach, the validity of the data as well as inconsistencies in the data, which in my opinion prohibit immediate publication.

i) Introduction, last paragraph and Results, first chapter.

Despite the clear presentation in Figure 1, there is no point-by-point description of the approach in the text or the legend to the figure. It is, hence, hard to follow and the approach raises some questions:

- why did the authors use the C_{β} distance in MD simulations

This is now clarified in the Figure 2 caption:

Figure 2. Design strategy for Bpy-Ala POP variants. (a) Filtering process used to select sites for Bpy LG pairs. **(b)** Distance and geometry requirements for Bpy LG placement were obtained by mapping the structure of a $\text{Zn}(\text{Bpy})_2$ (CCDC:756656) complex onto the Pfu POP structure

Point-by-point Response to Reviewers – Zubi et al.

(PDB ID: 5T88). A distance of $10.5 \pm 0.5 \text{ \AA}$ between the C-5 substituents on the two Bpy ligands and the C β atoms of the selected residues was used to ensure that Zn(BpyAla)₂ complex could be accommodated between the residues. (c) Locations of the final 27 LG pairs. The spheres represent the positions of the C β atoms of the selected residues on the β -propeller (black) and peptidase (grey) domains and labels represent residue numbers. Lines are drawn between residues in each pair, and the color of the line represents the change in distance between the residues in the open and closed POP conformations.

- why were MD simulations required when structures of both open and closed conformations of POP and luciferase exist

A structure of *Pfu* POP in the closed conformation does not exist (our structure, which was open, was the first reported), so MD simulations were required. Even though structures of Luc in different conformations do exist, we wanted to have a general protocol that would work for enzymes of interest. As noted above, we edited the sentence that clarifies this so that it specifically refers to conformationally dynamic systems:

As many proteins of interest could have a similarly large number of potential LG site pairs, or more, we developed a computationally guided approach for identifying sites for LG incorporation that could be applied to any protein that undergoes catalytically relevant conformation changes.

- what is meant by double variants

Consistent with the protein engineering literature, we use “mutant” to refer to an altered gene and “variant” to refer to an altered protein (no change was made to the manuscript). Double variants would therefore refer to protein variants that were designed to have two BpyAla residues in place of their native residues at positions that are specified.

- how does the metal-triggered switch work and related to this why does the Bpy-Ala side chains need to be located in close distance to each other

This switching mechanism is shown in Figure 1a. The following text was added to make this explicit in the text:

Structural analysis and MD simulations were used to identify residues that (i) are close enough in space to enable covalent bond formation between LG pairs in one enzyme conformational state (e.g., the closed conformation of POP), (ii) are solvent exposed to allow the proper chemistry to take place, (iii) undergo suitable changes in C β distance to prohibit bond formation in a different conformational state (e.g., the open conformation of POP), and (iv) are placed on opposite domains of the enzyme (Figure 2a). Because we were targeting metal regulation using genetically encoded bipyridinylalanine (BpyAla) residues,⁴⁷ the structure of a Zn(Bpy)₂ complex was analyzed to establish that a distance of approximately 10.5 \AA would allow for formation of a metal bis-BpyAla complex, M(BpyAla)₂, at the interface of the β -propeller and peptidase domains of POP (Figure 2b).

- how are the 180,000 potential double variants derived

Point-by-point Response to Reviewers – Zubi et al.

The 180,000 potential double variant number was an approximation derived by considering all possible double mutants of POP. In general, this is $n(n-1)/2$ where n is the number of residues in the protein. As POP is a 616 amino acid long protein, the number of double mutants is equal to $616(616-1)/2 = 189,420$ combinations. We have edited 180,000 to the exact number:

POP is a 72 kDa protein with 189,420 possible double variants.

ii) Results, chapter “HTE screening of POP switches”, 1st paragraph

The authors state that all POP variants were produced at full-length and without truncation products (legend to Supplementary Figure 2). However, supplementary Figure 2 does not prove this statement. In the autoradiogram in b) no protein marker was used to show that the molecular weight range is covered. In fact, some weak bands are visible for the majority of variants, which are missing for the wt POP.

We agree that analyzing the amount of full-length and truncated protein is crucial when analyzing the incorporation of noncanonical amino acids. To address this, we have edited Supplemental Figure 2b to feature a side-by-side comparison of the autoradiogram and the original dried SDS-PAGE gel which shows the relevant molecular weight ranges and markers (shown below). These gels and autoradiograms show that most smaller bands are common between all variants, which may suggest general protease activity within cell extracts. Bands suggesting truncation products, which generally are between 20 – 25 kDa or around ~59 kDa, do not appear to be strongly present. To clarify the presence of these smaller bands, we have edited the text in the figure legend of Supplemental Figure 2 to read:

Point-by-point Response to Reviewers – Zubi et al.

Supplementary Figure 2. BpyAla incorporation into POP.

A pair of BpyAla residues can be efficiently and accurately incorporated into all POP variants in CFPS. (a) All POP variants express solubly at yields comparable to WT POP, as measured by scintillation counting of proteins containing ^{14}C -leucine. Data represent mean \pm standard deviation from $n=3$ replicates. (b) Analysis of POP mobility in SDS-PAGE and in autoradiograms shows major products of identical size as WT POP. Bands further down on the gel may indicate the presence of smaller products from protease cleavage, but no major truncation product bands were observed. (c) Intact protein MS analysis of purified POP proteins shows masses in good agreement with incorporation of a pair of BpyAla residues. Theoretical masses are shown in Supplementary Table 4. Data in b are representative of $n=2$ independent experiments, and data in c were collected once.

In addition, the MS data do also not cover the molecular weight range of potential truncation fragments e.g., POP₂₁₆ ~ 25 kDa or POP₅₁₆ ~ 59 kDa. Showing the ratio of full-length versus truncated protein is crucial when the protein was not further purified.

The reviewer is correct in that the MS data as presented does not cover the range of potential truncation fragments. However, we note that no truncation products would be expected because POP variants were purified using a C-terminal StreptII tag, which would only purify full-length proteins. We have reanalyzed several of the variants as requested and show that this is the case. Only peaks above 20% relative ion counts were labelled, and no peaks of major truncation products were observed in any of the MS traces. Truncation product weights are included below.

Species	Truncation Product 1 (Da)	Truncation Product 2 (Da)
216/516	24994.35	59259.61
169/514	19617.19	59106.39
215/520	24880.24	59688.14
191/517	22183.14	59387.74
161/516	18808.3	59187.51

As the MS data would not be expected to show truncation products, we do not feel that this reanalyzed data would add to the manuscript and have not edited Supplemental Figure 2

Point-by-point Response to Reviewers – Zubi et al.

further. However, we have reiterated our purification approach in the text by adding additional detail in the Supplemental Methods:

POP Purification for MS Analysis:

CFPS reactions were scaled up to 75 μ L in a 15 mL conical tube and performed as previously described. POP containing a C-terminal StrepII tag was purified from the CFPS reaction using Strep-Tactin XT Spin Columns (IBA Life Sciences) according to the manufacturer's protocol. The C-terminal StrepII tag prevents purification of truncation products. POP purity was assessed by SDS-PAGE. Proteins were quantified by Nanodrop using extinction coefficients and molecular weights calculated by ExPasy ProtParam (<https://web.expasy.org/protparam/>).

Intact Protein Mass Spectrometry

Purified proteins were buffer exchanged into PBS (1.37 mM NaCl, 27 mM KCl, 100 mM Na₂HPO₄, 18 mM KH₂PO₄, pH 7.4) using Amicon Ultra-0.5 10 kDa MWCO Centrifugal filters. Purified proteins in PBS were then analyzed by LC-MS, as described in previous publications.²⁴ m/z data was deconvoluted between a mass range of 70,000 – 75,000 Da, as only full-length protein was expected to be purified.

Interestingly, as is apparent from gels of POP variants purified by IMAC using Ni-NTA resin, (e.g. samples used for data collection presented in *Functional characterization of a BpyAla POP Switch*), truncated protein is observed. This is different than what was seen with proteins purified for HT analysis (e.g. POP purified by Strep-Tactin XT Spin Columns), which contained no significant amount of truncated protein. We have previously found that truncated POP has affinity for Ni-NTA resin even if it doesn't contain a His₆ tag. This phenomenon has been known to our lab for several years (first discovered by those working on developing POP artificial metalloenzymes) and has been validated by several students.

iii) Results, chapter "HTE screening of POP switches", 1st paragraph

A high-throughput screening system was set up to "enable screening and characterization of hundreds of enzyme variants and reactions in less than a day." However, the authors have produced 27 and not hundreds of variants to test.

We thank the reviewer for the feedback. The reviewer is correct in stating that hundreds of protein variants were not tested within this work. Our intent was to convey that we could test hundreds of candidates with the workflow if there is a suitable activity assay. Indeed, cell-free systems have been used to assess even millions of reactions (10.1038/s41467-020-15798-5). However, as hundreds of variants were not tested in this work, we have edited the sentence to read:

The HTE workflow is purification-free and could allow for analysis of hundreds of reactions in less than a day, with the potential to screen large numbers of variants with a suitable functional assay.

Moreover, with regard to the ambition to apply the approach to any protein, it will be difficult or hardly possibly for many other enzymes to set up such a HTE screening since not all reactions allow a spectrophotometric detection. It should be discussed in the discussion section how this step can be adapted for other proteins e.g., for those without the possibility to monitor activity spectrophotometrically.

Point-by-point Response to Reviewers – Zubi et al.

We thank the reviewer for highlighting this point of clarification and agree that enzyme activity cannot always be directly measured spectrophotometrically, as was done for POP and *p*-nitroaniline. This, of course, is a challenge for any enzyme and not specific to our method for controlling enzyme activity. That said, we note that our approach is amenable to a variety of other detection methods, including fluorescence and luminescence (as was done for Pluc), that are compatible with modern instrumentation such as plate readers. In addition, any molecule that can be measured spectrophotometrically, such as NADH or other molecules associated with enzymes, could be used to monitor activity. We now discuss these challenges and opportunities in the discussion section as the reviewer requests. Specifically, we write:

While this workflow is limited to enzymes that have substrates, products, or cofactors compatible with high-throughput measurement such as those with optical, fluorescent, or luminescent properties, it nevertheless facilitates screening of many types of enzymes. For example, the activity of any enzyme that oxidizes or reduces NADH could be measured in real-time using this workflow. This HTE approach enabled us to engineer a switch, comprising a pair of genetically encoded BpyAla residues that are strategically incorporated so that reversible metal coordination would bias enzyme conformational states. The Bpy LG was incorporated into dozens of sites in two structurally distinct enzymes, Pfu POP and Pluc, with high yields and only modest impact on the activity of functional variants in the absence of metal, highlighting its functional-orthogonality and potential utility for proteins that cannot accommodate large conformationally responsive domains or the introduction of new catalytic sites.

iv) Results, chapters “HTE screening of POP switches” and “Functional characterization of a BbyAla POP switch”, throughout

Of the 27 tested variants, the authors have identified three, POP_{167/517}, POP_{169/510}, and POP_{159/517}, with Ni(ii)-mediated inhibition of POP activity (2nd paragraph, line7). Unfortunately, the results are not presented consistently for these three variants:

- Figure 3a only highlights POP_{167/517}, and POP_{159/517} but not POP_{169/510}
 - the decrease factor in line 10 is also only given for POP_{167/517}, and POP_{159/517}
 - SDS-PAGE gels for purified POP is only shown for POP_{167/517}, although steady-state kinetics of purified POP_{159/517} (Figure 3c; Table 1) and POP_{169/510} (Supplementary Figure 6) are shown as well
 - the steady-state kinetics of POP_{169/510} is only shown in Supplementary Figure 6 without any mentioning thereof in the main text
 - in Supplementary Figure 6 a fourth variant appears, POP_{169/512}, which is also not mentioned in the main text
 - two controls POP₁₆₇ and POP₅₁₇ for POP_{167/517} were analyzed in steady-state kinetics, however, appropriate control variants for the other two variants POP_{159/517} and POP_{169/510} are missing
 - further characterizations after steady-state kinetics were only performed for POP_{167/517}
- The authors should refrain from jumping between four, three, two or only one variant by supplementing the missing data and clearly explaining why for the full characterization only POP_{167/517} is regarded (I suspect because it showed the highest control factor).

We thank the reviewer for pointing out this inconsistency in how we selected variants to describe the text. We have now only highlighted variants discussed in text, referencing all else in SI. Changes include:

Point-by-point Response to Reviewers – Zubi et al.

- (1) Focused results entirely on five variants, four of which were deactivated (167/517, 169/510, 159/517, and 169/512) and one of which was activated (167/513) to different extents in the main text:

We first used this workflow to screen POP variants for Ni(II) inhibition in the presence of 0 – 1,000-fold molar excess of Ni(II) relative to enzyme (Figure 3a, 0.1 μ M enzyme). Control reactions indicated that blank CFPS extracts and 2TAG-sfGFP were inactive, and no background inhibition of WT POP occurred at any Ni(II) concentration. On the other hand, all but five POP variants displayed activity, and ten showed enzymatic rates >50% that of WT. Of these ten, six showed dose-dependent inhibition by Ni(II), which we defined as less than 50% apo activity at the highest concentration of metal tested (Supplementary Table 1). The top four variants with the greatest degree of inhibition, POP_{167/517}, POP_{169/510}, POP_{159/517}, and POP_{169/512}, displayed nearly complete inhibition (\leq 5% apo activity) at higher concentrations of Ni(II) while maintaining > 75% of WT activity in the absence of metal (Figure 3a, shaded graphs, red text). At intermediate concentrations of Ni(II), these four variants exhibited 2.8-14 fold decreased rates. Interestingly, POP_{167/513} displayed >50% increase in activity in the presence of Ni(II) relative to the apo protein at the highest concentration of metal tested (Figure 3a, shaded graph, blue text). The proximity of 167/517 and 167/513 in primary sequence space highlights how subtle changes in protein structure can have significant impacts on function (i.e., metal inhibition and activation, respectively) and the relatively unexplored space in understanding conformational effects on protein function.

We next characterized POP_{X/Y} responses to different divalent metal cations, including Cu(II), Co(II), Zn(II), and Fe(II) (Figure 3b and Supplementary Figure 3). While WT POP did not respond to these metals, POP_{169/510}, POP_{159/517}, POP_{167/517}, and POP_{169/512} were inhibited by Cu(II), Co(II), and Zn(II) in a dose-dependent manner, but only the latter two were inhibited by Fe(II). POP_{167/513}, which was activated by Ni(II), was also activated by Cu(II) and Co(II) but not Zn(II) or Fe(II) (Figure 3b, Supplemental Figure 3). These differences in extent of activation and inhibition may reflect different preferred coordination geometries of the M(Bpy)_n complexes or the relative binding affinities of the Bpy LG for the different metals.⁵¹

The reversibility of POP variant inhibition/activation was examined by adding a competitive chelator, ethylenediaminetetraacetic acid (EDTA), to reactions conducted using the workflow outlined above (Figure 3b, Supplementary Figure 4). EDTA formation constants for complexation with Ni(II), Cu(II), Co(II), Zn(II), and Fe(II) are $\sim 10^{14}$ - 10^{18} (Fe < Co < Zn < Ni < Cu), which exceed values for M(Bpy)₂ complex formation and thus should enable reversibility by competition.⁵² All POP variants were incubated with 1000-fold excess metal, exposed to 2 mM EDTA, and assayed to measure activity. Nearly all enzymes tested with Ni(II), Cu(II), Co(II), or Zn(II) displayed near-quantitative recovery of activity upon addition of EDTA (Figure 3b). Recovery of activity was not observed following addition of EDTA to variants treated with Fe(II), perhaps due to the relatively low affinity of EDTA for Fe(II) relative to the other metals tested. Activation of POP_{167/513} was also found to be reversible by EDTA-based chelation. These results highlight how our HTE workflow facilitated rapid identification of POP variants that displayed reversible catalytic responses to divalent metals in a manner consistent with the formation of a M(BpyAla)₂ complex involving Bpy LGs.

Point-by-point Response to Reviewers – Zubi et al.

We next studied the catalytic properties of $POP_{167/517}$, $POP_{169/510}$, $POP_{159/517}$, $POP_{169/512}$ and $POP_{167/513}$, which exhibited the highest degrees of reversible inhibition or activation by metal salts in our HTE workflow (Figure 3c, Supplementary Figure 6).

- (2) Highlighted and included fold improvement numbers on 169/510, 169/512, and 167/513, in Figure 3a and Supplemental Figure 3 so that all five featured variants are presented in the same way.
- (3) Removed mention of 156/513 from the text to focus on the five variants noted above
- (4) Added data for 169/512 to Figure 3b so that all five featured variants are presented in the same way in Figure 3b.
- (5) Because there is not room or need to include steady state kinetics for all five variants in the manuscript, we removed 159/517 from Figure 3c so that this figure only shows a single example of the comparison in steady state kinetics between the best deactivated variant ($POP_{167/517}$) and WT.
- (6) Additionally, as we discuss more in detail below, we have added data for activity in the presence of Fe(II) in Figure 3b.

Figure 3. Kinetic analysis of $POP_{X/Y}$ variants in response to divalent metals. (a,b) The activity of $POP_{X/Y}$ variants from CFPS was assessed on Z-Ala-Pro-pNA. (a) Reaction rates (AU/min) at increasing Ni(II) concentrations (0-100 μ M). Inhibited variants are highlighted in red. The fold-changes between 0 and 10 μ M of Ni(II) for $POP_{167/517}$, $POP_{167/513}$, $POP_{169/512}$, $POP_{169/510}$, and $POP_{159/517}$ are noted. (b) The reversibility of metal-dependent activity upon addition of EDTA shows near-quantitative recovery for Ni(II), Cu(II), Co(II), and Zn(II), while activity is not recovered for changes by Fe(II). Select variants from Figure 3a are shown. (c) Steady-state kinetic assays performed at 85 $^{\circ}$ C using purified enzymes (\sim 20 nM) in the presence of either 1 mM EDTA (black) or 5 μ M NiCl₂ (green). Initial rates (μ M/sec) are plotted versus substrate concentration (mM), and data was fit, when appropriate, with the Michaelis-Menten equation. The fold-change between reaction rate at 1 mM substrate is shown. Each data point represents the average of 3 replicates and error bars represent standard deviations.

Point-by-point Response to Reviewers – Zubi *et al.*

- (7) Steady state kinetics for the other four variants noted above are mentioned in the text and data for these are provided in the SI. Additionally, kinetics for the 1:1 mixture of POP₁₆₇ and POP₅₁₇ are shown in this figure and table as was suggested by reviewer 2. Supplementary Figure 5 was edited to reflect this, and Supplementary Table 2 was added as follows:

Point-by-point Response to Reviewers – Zubi et al.

Point-by-point Response to Reviewers – Zubi et al.

Supplementary Figure 5. Kinetic traces for BpyAla POP variants inhibited by Ni(II). Steady-state kinetic assays were performed at 85 °C with purified enzymes (~10-30 nM) in either the presence of 1 mM EDTA (black) or 5 μ M NiCl₂ (green). Initial rates of reaction (μ M/sec) are plotted versus substrate concentration (mM) and data was fit, when appropriate, with the Michaelis-Menten equation (Equation 2). The fold-change (EDTA vs. Ni²⁺) between reaction rates at 1 mM substrate is shown. Each data point represents the average of 3 replicates (n=3) and error bars represent standard deviations.

Supplementary Table 2. Steady-state kinetic parameters for selected POP variants.

^a Variant	K_M (μ M)		k_{cat} (sec^{-1})	
	+ EDTA	+ NiCl ₂	+ EDTA	+ NiCl ₂
POP _{159/517}	265 \pm 38	523 \pm 73	139 \pm 8	57 \pm 4
POP _{169/510}	409 \pm 20	NA	154 \pm 4	NA
POP _{169/512}	428 \pm 36	NA	91 \pm 4	NA
POP _{167/513}	854 \pm 102	661 \pm 43	219 \pm 16	259 \pm 9
(1:1) POP ₁₆₇ and POP ₅₁₇	340 \pm 19	317 \pm 22	275 \pm 7	225 \pm 7

^aReactions were conducted in triplicate using 0-1 mM Z-Ala-Pro-pNA and 20-21 nM enzyme in 10% v/v DMSO/30 mM HEPES (pH 7.4) containing 0.8 M NaCl at 85 °C for 1 minute. Rates were determined by changes in absorbance over time at 410 nm using a calculated molar extinction coefficient for pNA (7,126 M⁻¹ cm⁻¹). Kinetic parameters were determined by the non-linear regression function in OriginPro using the Michaelis-Menten equation (Equation 2).

Additionally, we have moved kinetic parameters for variants POP₁₆₇ and POP₅₁₇ to Table 1 in the main text as suggested below, under **Minor Points**. Additionally, errors for determination of kinetic parameters from fitting with the Michaelis-Menten equation have been added to the table.

- (8) Added SDS-PAGE gel of variants 159/517, 169/510, 169/512, and 167/513 to Supplementary Figure 6 and edited the figure caption as follows:

Point-by-point Response to Reviewers – Zubi et al.

Supplementary Figure 6. Representative SDS-PAGE gel of purified POP variants. SDS-PAGE analysis of (top) POP_{WT}, POP₁₆₇, POP₅₁₇, and POP_{167/517} before and after SEC purification (SEC purification was performed in the presence of excess EDTA) and (below) POP_{159/517}, POP_{169/510}, POP_{169/512}, and POP_{167/513} following IMAC purification. PageRuler™ Prestained Protein Ladder or PageRuler™ Plus Prestained Protein Ladder (Thermo Fisher) were used for reference. 4 µg of each protein sample was loaded onto the gel. In addition to a band corresponding to the MW of the desired full-length protein, we also observe a small amount of the truncated protein, that forms because of competition between release factor 1 mediated (RF1) termination of translation and BpyAla incorporation at the amber stop codon¹. The MW of the variants and relevant truncated forms are as follows: POP_{WT} (71,888.78 Da); POP₁₆₇ full-length (71,999.93 Da); POP₁₆₇ truncated (19,405.80 Da); POP₅₁₇ full-length (72,014.90 Da); POP₅₁₇ truncated (59,306.42 Da); POP_{167/517} full-length (72,126.04 Da); POP_{167/517} 517-truncated (59,330.50 Da); POP_{159/517} full-length (72,111.98 Da); POP₁₅₉ truncated (18,550.84 Da); POP_{159/517} 517-truncated (59,403.50 Da); POP_{169/510} full-length (72,105.03 Da); POP₁₆₉ truncated (19,617.02 Da); POP_{169/510} 510-truncated (58,635.63 Da); POP_{169/512} full-length (72,129.02 Da); POP_{169/512} 512-truncated (58,900.95 Da); POP_{167/513} full-length (72,062.01 Da); POP_{167/513} 513-truncated (58,997.11 Da).

Point-by-point Response to Reviewers – Zubi et al.

- (9) Clarified in the main text that further characterization of 167/517 (including kinetic analysis of the single-point mutants e.g. POP₁₆₇ and POP₅₁₇) was based on the fact that this variant that had best control factor:

Particularly notable is POP_{167/517}, which displayed a 17-fold change in activity in response to Ni(II) (Figure 3c).

Given the fact that POP_{167/517} provided the highest level of control of any of the variants examined, further characterization of this variant was pursued.

v) Results, chapter “Functional characterization of a BpyAla POP switch”, 1st paragraph
In steady-state kinetics of POP_{167/517} the authors do not give the K_M and k_{cat} in presence of Ni(II) without further statement. The graph in Figure 3c is quite small so it is not apparent for the reader why the constants could not be determined. I suspect the K_M value is too high to be resolved. However, this should be made clear and also, if this is the case, why they did not increase the substrate concentration range to determine the constants. If increasing substrate concentration has no practical limitations, I strongly suggest to re-measure the steady-state kinetics.

We thank the review for this feedback. Indeed, the K_M and k_{cat} in the presence of Ni (II) could not be accurately determined as saturating conditions of substrate were not approached. Further increasing substrate concentration was not possible due to the low solubility of the substrate. We have added in a sentence to that paragraph to clarify this point:

k_{cat} and K_M could not be calculated for POP_{167/517} in the presence of Ni(II) because saturating substrate concentrations under these conditions were beyond the solubility limit of the substrate.

vi) Results, chapter “Functional characterization of a BpyAla POP switch”, last paragraph
I understand why the authors used iron to visualize the metal-BpyAla linkage in the UV/Vis. However, to correctly conclude that the formation of the linkage is also the cause for a drop in activity, the authors must show that inhibition of POP also works with Fe(II). They have shown successfully that inhibition works with Ni(II), Cu(II), Co(II) and Zn(II) in Supplementary Figure 4 but Fe(II) data are missing. The data should be provided before publication.

We understand the reviewer’s concern about missing data for kinetic inhibition of variants of interest by Fe(II) and we appreciate the opportunity to add additional data that we believe strengthens our paper. We have conducted additional experiments to screen how Fe(II) may affect enzyme behavior in the panel of enzyme variants. (Figure 3b, Supplementary Figure 3). We find that two of our best-performing variants, POP_{167/517} and POP_{169/512} meet our criteria for < 50% apo activity at the highest concentration of Fe(II) tested in this experiment. As before, these metal concentrations were not observed to inhibit the wild-type enzyme. **We note that POP_{167/517} was the variant that was taken for further characterization, and that it was responsive to Fe(II) in a manner similar to other metals.** We find that response to Fe(II) is less common throughout the panel of variants tested. Fe(II) has a significantly weaker constant of formation for M(Bpy)² complexes at $K \sim 10^8$ compared to Ni, which is on the order of 10^{14} . This weaker constant of formation may explain the observation that Fe(II) is less effective at inhibiting POP variants in this experiment, particularly at intermediate doses of Fe(II).

Point-by-point Response to Reviewers – Zubi et al.

We have also conducted experiments to test for reversibility of Fe(II)-based inhibition of the panel of enzyme variants (Supplementary Figure 4). Interestingly, we found that inhibition by Fe(II) is not reversible under the same conditions used for Ni(II), Cu(II), Co(II), and Zn(II). This is supported by previous literature, in which phenanthroline was necessary for Fe(II) removal from BpyAla complexes ([10.1073/pnas.1600188113](https://doi.org/10.1073/pnas.1600188113)).

Supplementary Figure 3. High-throughput screening of POP activity in response to different divalent metals.

High-throughput experiments for POP rates in response to a range of (a) Cu, (b) Co, (c) Zn, and (d) Fe concentrations show that M(II) ions can inhibit and activate select POP variants in a dose dependent manner. Data represent mean \pm standard deviation from $n=3$ replicates and were replicated in two independent experiments.

Point-by-point Response to Reviewers – Zubi et al.

Supplementary Figure 4. High-throughput screening of reversibility of metal-dependent activity changes in POP. High-throughput screening of reversibility shows that addition of EDTA to POP enzymes that are pre-treated with (a) Ni, (b) Cu, (c) Co, (d) Zn, and (e) Fe can restore apo-enzyme activity. Data represent mean \pm standard deviation from $n=3$ independent replicates and were replicated in two independent experiments.

We have added the following text to address the Fe(II) data:

While WT POP did not respond to these metals, POP_{169/510}, POP_{159/517}, POP_{167/517}, and POP_{169/512} were inhibited by Cu(II), Co(II), and Zn(II) in a dose-dependent manner, but only the latter two were inhibited by Fe(II). POP_{167/513}, which was activated by Ni(II), was also activated by Cu(II) and Co(II) but not Zn(II) or Fe(II) (Figure 3b, Supplemental Figure 3). These differences in extent of activation and inhibition may reflect different preferred coordination geometries of the $M(\text{Bpy})_n$ complexes or the relative binding affinities of the Bpy LG for the different metals⁵².

Recovery of activity was not observed following addition of EDTA to variants treated with Fe(II), perhaps due to the relatively low affinity of EDTA for Fe(II) relative to the other metals tested

Along with data showing inhibition of POP_{167/517} by Fe(II) (Figure 3b), these data suggest that bidentate metal coordination by two Bpy LGs to generate a $M(\text{II})(\text{BpyAla})_2$ linkage as shown in Figure 4e favors the closed conformation of POP_{167/517} and prohibits catalytic turnover.

- vii) SI, chapter “Physical Characterization of POP Variants – Circular Dichroism”, p.45
The integrity of the secondary structure of POP variants in absence and presence of Fe(II) cannot be properly validated. The representation should focus on the far UV range of ~185 to ~260 nm. From what is shown, two issues should be addressed:
- The noise below 200 nm seems to be significant and hence no clear transition at ~190 nm indicative of the $\pi \rightarrow \pi^*$ transition of peptide bonds is visible. Since the protein is stored in H₂O no buffer components cause this effect. The authors should check the High Tension voltage traces. If these indicate overloading, the CD spectra should be remeasured with lower protein concentration or in a thinner cuvette to prove the structural integrity.
 - The signal strength at ~220 nm differs significantly between the samples, which might be indicative of a decrease in secondary structure potentially caused by protein degradation. It

Point-by-point Response to Reviewers – Zubi et al.

might also point to variations in protein concentration. The authors should check this.

We agree that this needed to be resolved and thank the reviewer for suggesting we repeat these experiments at lower concentrations. As such, CD were collected at 10 nM for POP variants WT, 167, 517, and 167/517 in the absence or presence of Fe(II) using the same exact protein stock solutions (stored at -80 °C) used to collect the older data. The voltage traces for these samples were more reasonable and did not indicate any overloading as was the case for data collected with 5 μ M protein. Moreover, these spectra closely match previously collected CD of *Pfu* POP (<https://www.nature.com/articles/ncomms8789>) and also resemble those for a related POP from porcine brain (<https://www.sciencedirect.com/science/article/pii/S2451945619302715?via%3Dihub>).

We have revised Supplementary Figure 22 (now Supplementary Fig. 21) with these data as shown below:

Supplementary Figure 21. CD of various dilute POP samples.

CD spectra were collected of POP_{WT}, POP₁₆₇, POP₅₁₇, and POP_{167/517} (all at 10 nM) in the absence or presence of Fe(II). At the higher concentrations (50 μ M) necessary to visualize certain features associated with the metalloprotein (i.e. Cotton effects near 300 nm), the signal was too high to clearly observe secondary structural elements of the protein present in the far-UV region, thus we measured spectra at lower concentrations as well. A deep peak around 210 nm, characteristic of folded POP was observed.

In the original Supplementary Figure 22, there were differences between signal at 220 nm. As was suggested by the reviewer, this can be attributed to differences in protein concentration or small changes in secondary structure content due to protein degradation. However, upon repeating this assay at lower concentrations (10 nM vs. 5 μ M), the differences were not as large, and perhaps were exaggerated when assays were conducted beyond the linear response

Point-by-point Response to Reviewers – Zubi et al.

region at higher concentrations (i.e. 5 μ M) as would be evident by large High Tension Voltage values.

Minor points

Introduction, last paragraph: "...used MD simulations to identify all residue pairs that undergo significant changes... during catalysis."

- Rephrase. To my knowledge the actual catalysis event cannot be simulated with MDs.

Updated as requested:

Our framework (Figure 1b) used molecular dynamics (MD) simulations to identify all residue pairs that undergo significant changes in C_{β} distance between POP conformations.

Introduction, last paragraph: "...to select functional candidates for analysis."

- Clarify what is meant by "functional candidate".

Updated as requested:

Enzyme variants containing LG pairs at candidate sites were then designed and rapidly prototyped using cell-free protein synthesis (CFPS)⁴⁴ and high-throughput experimentation (HTE) to select active enzyme variants with metal-dependent activity for analysis.

Results, chapter "Model selection and design of POP switches", 1st paragraph

- The authors talk once of "domains" and then again of "subunits". Clearly the β -propeller and the peptidase parts of POP are domains since they are located on the same protein chain. The term subunits is used for two separate protein chains. Replace "subunits" with "domains".

Updated as requested.

Results, chapter "Model selection and design of POP switches", Figure 2

- which PDB was used for the representation of POP?

Updated as requested:

(b) Distance and geometry requirements for Bpy LG placement were obtained by mapping the structure of a $Zn(Bpy)_2$ (CCDC:756656) complex onto the Pfu POP structure (PDB ID: 5T88). A distance of 10.5 ± 0.5 Å between the C-5 substituents on the two Bpy ligands and the C_{β} atoms of the selected residues was used to ensure that $Zn(BpyAla)_2$ complex could be accommodated between the residues.

- in a) "Placement between domains" has not been mentioned as a criterium in the main text

Updated as requested:

Structural analysis and MD simulations were used to identify residues that (i) are close enough in space to enable covalent bond formation between LG pairs in one enzyme conformational state (e.g. the closed conformation of POP), (ii) are solvent exposed to allow the proper chemistry to take place, (iii) undergo suitable changes in C_{β} distance to

Point-by-point Response to Reviewers – Zubi et al.

prohibit bond formation in a different conformational state (e.g. the open conformation of POP), and (iv) are placed on opposite domains of the enzyme (Figure 2a).

- in b) Is the Zn(Bby)₂ Ligand correctly oriented? Further guidance including atom-wise coloring and orientation similar to the Bby-Ala in Figure 1a would help to understand how the 10.4 Å distance was derived.

Updated as requested:

Figure 2b was updated so that: the protein was oriented in a similar way as presented in Figure 1a; the Bpy ligands on the Zn complex were colored crimson red; the C β atoms of the residues on which the Zn complex is mapped were colored crimson; the metal of the Zn complex was colored gray; and finally, the other cis ligands were colored gold to distinguish them from the Bpy ligands. See above for the updated figure caption.

Results, chapter “Model selection and design of POP switches”, 2nd paragraph
“(i) ... in one enzyme conformational state,” - in which state? Closed or open?
“(ii) undergo suitable changes in C β distance to prohibit bond formation a different conformational state.” - Clarify this point. I understand this criterium that metal binding should only occur in one conformation but not the other and that hence the two Bby-Ala positions need to be close in one conformation but distant in the other.

We have parenthetically noted the relevant states in the case of POP. Different states may be relevant to different enzymes where “open” or “closed” are not appropriate descriptors:

Structural analysis and MD simulations were used to identify residues that (i) are close enough in space to enable covalent bond formation between LG pairs in one enzyme

Point-by-point Response to Reviewers – Zubi et al.

conformational state (e.g. the closed conformation of POP), (ii) are solvent exposed to allow the proper chemistry to take place, (iii) undergo suitable changes in C_{β} distance to prohibit bond formation in a different conformational state (e.g. the open conformation of POP), and (iv) are placed on opposite domains of the enzyme (Figure 2a).

Results, chapter “HTE screening of POP switches”, 1st paragraph, “... at an average yield of 1098 +/- 124 $\mu\text{g/mL}$,...”

- Define $\mu\text{g/mL}$. Do the authors mean the yielded concentration or do they mean 124 μg protein yielded from 1 mL expression volume?

In the CFPS literature, yields refer to the concentration of protein measured within the reaction. A yield of 1000 $\mu\text{g/mL}$ means that 1000 μg of protein would be synthesized in a 1 mL reaction. As detailed in the SI, our reactions in this paper were performed at 15 μL scale. To clarify this point, we have edited the appropriate section to be more clear:

Second, we synthesized the entire panel of POP variants ($\text{POP}_{X/Y}$, where X and Y are the sites of BpyAla incorporation) at an average yield of 1098 +/-124 $\mu\text{g/mL}$ within a 15 μL CFPS reaction, comparable to wild-type (WT) protein (Supplementary Figure 2).

Results, chapter “HTE screening of POP switches”, 2nd paragraph, “... understanding allosteric effects on protein function.”

- Replace “allosteric” with “conformational”.

Corrected as requested.

Results, Supplementary Figure 3&4

- Highlight the three variants $\text{POP}_{167/517}$, $\text{POP}_{169/510}$ and $\text{POP}_{159/517}$ as in Figure 3a

Corrected as requested. All variants boxed in Figure 3a and highlighted in Figure 3b are now consistently labelled, with fold-activation and fold-inhibition listed in Supplemental Figure 3 and the same variants boxed in Supplemental Figure 4.

Results, chapter “HTE screening of POP switches”, 3rd paragraph, “... showing the weakest degree of inhibition or activation...”

- Give quantitative numbers as in Figure 3a in the main text and Supplementary Figure 3.

Numbers have been added to all variants of choice. We provide quantitative numbers for fold-change for activation and inhibition for the variants that we focus on. We chose to focus on $\text{POP}_{167/517}$, $\text{POP}_{169/510}$, $\text{POP}_{159/517}$, and $\text{POP}_{169/512}$ for the inhibiting variants, and now provide fold activation for $\text{POP}_{167/513}$. The criteria for the selection of the inhibiting variants is now clearly described in the text and is as previously mentioned in Major Points (3) (iv).

Results, chapter “Functional characterization of a BbyAla POP switch”, Supplementary Figure 5

- Full-length and truncated protein should be highlighted in the figure or the molecular weight given in the legend.

Corrected as requested in what was Supplementary Figure 5 (now Supplementary Figure 6)

Results, chapter “Functional characterization of a BbyAla POP switch”, Figure 3c

- What is meant by the labels “167N517V” and “159K517V”?

Point-by-point Response to Reviewers – Zubi et al.

This was a result of an inconsistency with labelling of BpyAla variants. We thank the reviewer for pointing this out. We have corrected as requested.

Results, chapter “Functional characterization of a BbyAla POP switch”, Table 1
- Provide the standard error of the Michaelis-Menten fit as calculated by Origin.

Corrected as requested.

Results, chapter “Extension to luciferase chemical switches”, Figure 5
- Why does luciferase show a substrate inhibition effect? Is this known for the enzyme? Please clarify in the main text.

It has been previously shown by others that the kinetics of luciferase can be affected by varying the concentrations of its substrate including D-luciferin and ATP. Byproducts of the reaction like oxyluciferin and L-AMP can inhibit the enzyme (<https://www.sciencedirect.com/science/article/pii/S1074552110001973?via%3Dihub>). This could be the reason we observe lower activity at increasing substrate concentrations. As we do not know the exact reason for this behavior, we have modified the text and now include this reference in the caption for Table 1:

^bReactions were conducted by mixing 1.1 μ M enzyme (preincubated with an equal volume of 1 mM Ni (II) when applicable) with 0 – 0.75 mM D-luciferin in DMSO (5% v/v DMSO total) in 12.5 mM HEPES (pH 7.8), 5 mM MgSO₄, and 1 mM ATP. Luminescence was read for 5 min at room temperature immediately after mixing. The maximum RLU values at each D-luciferin concentration, which describe the “glow” phase of the Pluc reaction mechanism,⁵⁴ were fit to a Michaelis-Menten equation that accounts for inhibition observed at increasing substrate concentrations. This inhibition could be the result of competitive concentrations of inhibitory byproducts⁵⁵. Max Rates describe apparent V_{max} of Pluc in the glow phase. Errors for each parameter as calculated in OriginPro from triplicate reactions are provided.

SI, chapter “MD simulations of POP” and “MD simulations of Luciferase” p. 26
- For reproducibility reasons please provide the PDB-IDs for POP and luciferase structures used in MD simulations

POP:5T88

Luciferase: WT (4G36) and cross-linked (4G37)

- Clarify in more detail how the BpyAla side chains were introduced.

In VMD, once the structure and connectivity of the protein and the desired side chain were settled, you can mutate the side chain as the amino acid residue.

- What is meant by “cross-linked luciferase”? Please explain.

The cross-linked luciferase referred to in the Supplementary Information comes from (<https://pubs.acs.org/doi/10.1021/bi300934s>), which used a chemical cross-linker to trap Pluc in a conformation that was incompetent for the adenylation reaction but competent for the oxidative reaction. We used this crystal structure to design luciferase variants that were trapped

Point-by-point Response to Reviewers – Zubi et al.

in the oxidative-competent conformation, which would render Pluc sensitive to metals by preventing the adenylation reaction from occurring. We treated cross-linked structure as another confirmation of the wild-type (by mutating the residues from cross-linked one to all wild-type one, excluding the ones that are highly conserved, while keeping the backbone structure as the cross-linked one. The initial structures of both have to go through MD simulations).

SI, chapter “Cell extract preparation” p. 31

- pEVOL vectors contain constitutive promoters for aaRS1 and tRNA as well as an araC induction system for aaRS2. However, here IPTG was used for induction. Does pEVOL-BpyRS differ from other pEVOLs?

We appreciate the opportunity to clarify this point. We used IPTG to induce T7 RNA polymerase expression from the *E. coli* strain used in this study, 759.T7 ([https://www.cell.com/cell-chemical-biology/pdfExtended/S2451-9456\(19\)30353-8](https://www.cell.com/cell-chemical-biology/pdfExtended/S2451-9456(19)30353-8)). This strain contains a T7 RNA polymerase gene downstream of an Lpp5 promoter and lac operator. pEVOL-BpyRS contains the traditional architecture of constitutive promoters for aaRS1 and tRNA, as well as pBAD induction system for aaRS2. The plasmid merely served to provide tRNA for the CFPS reaction.

SI, chapter “Steady-State Kinetic Assays” p.39-41

- “For steady-state kinetic assays with variants POP167/513, POP169/510, POP169/512, and POP167/516, protein samples ...” Three of the variants have never been mentioned in the main text, however, POP159/517 is missing.

- Why were only POP wt, POP167, POP517, and POP167/517 purified after EDTA treatment by SEC? Could diafiltration for the other variants leave traces of EDTA, which might have inhibited POP activity in the kinetic assays?

We thank the reviewer for noticing this oversight in the SI. We also purified POP159/517 in the same manner as those other variants mentioned here and have edited the SI to reflect that.

We further purified these variants of interest with SEC for the purposes of intact ESI-MS and crystallization efforts (the latter of which has failed thus far) where a secondary purification step was desirable. Diafiltration was performed extensively for the other variants so that the theoretical concentration of EDTA was in the nM regime. This small amount of EDTA should not have a significant effect on the kinetic assays which were either performed with an excess of EDTA (1 mM) or Ni(II) (5 μ M).

- Why was POP transferred to pure H₂O. Most proteins require a buffer system for stability, why not POP?

We have extensive experience working with *Pfu* POP and have found that it is thermostable and tolerates storage in MQ H₂O.

- Why does Supplementary Figure 19 show exemplary kinetic traces for POP167/513 which is never mentioned in the main text?

We have corrected the main text (see above) so that this activating variant is referenced to, and data presentation was made consistent with other highlighted variants.

- The data for POP167, POP517 and POP169/510 in Supplementary Table 6 should be transferred to Table 1 or at least referred to in the main text.

Point-by-point Response to Reviewers – Zubi et al.

We have moved the data for POP167 and POP517 to the main text Table 1 and have moved data for POP169/510 to Supplementary Table 2 in the SI to be consistent with other variants as noted above.

- Data for POP169/512 and POP167/513 in Supplementary Table 6 are irrelevant because they have never been mentioned in the main text.

We have addressed this by mentioning these variants in the main text along with other variants of interest and editing Figures and Tables to reflect this (see above).

Reviewer 2

Comments:

Zubi, et al demonstrated that the formation of M(bpy)₂ species can endow a genetically encoded chemical switch to regulate the catalytic activities. They conducted structural analysis and MD simulation to assess the desirable positions to incorporate a pair of bpy-Ala residues. Consequently, 24 POP variants were prepared by cell-free protein expression systems, and at least 12 ones are responsive to the first-row transition divalent metal ions, such as Co, Ni, Cu, and Zn. In addition, the proof of concept was applied to luciferase, expanding the scope of systems.

1) One of the general prerequisites for any chemical switch would be not to perturb the native catalytic activity and protein stability. Otherwise, it would be no longer useful. In this regard, the catalytic activities of Pluc variants were substantially perturbed from the wild-type, yielding 25% or 50% of the V_{max} even in the absence of Ni ion. Can authors speculate how such a significant loss of activities results from the incorporation of bpy-Ala residue? Perhaps, are bpy-Ala residues incorporated in the vicinity to the catalytically essential residues? Based on these results, can authors improve the designing scheme to propose the metal-dependent switches?

In general, reversible switching of enzyme activity almost always involves a reduction in the activity of the WT enzyme. In an earlier study, we provided simulations that suggest that the catalytically essential H592 residue hydrogen bonds with a number of residues, some of which were targeted in the current study, as it transits from its orientation in the open and closed forms of the enzyme (<https://pubs.acs.org/doi/abs/10.1021/acs.biochem.9b00031>). Disrupting those interactions could cause problems with catalysis. Given that this is only speculation, as the reviewer suggested, and given the complexity of those interactions and their relevance to catalysis even if the speculation were to prove accurate, we think the best approach to correcting this issue would be via directed evolution (and this would also apply for luciferase, where we saw larger decreases in apo activity upon incorporation of BpyAla). Screening for activity in the presence and absence of metal ions would allow for identification of enzymes whose activity is not compromised relative to parent in the absence of metal and improved relative to parent in the presence of metal. This is an approach we plan to pursue in due course.

2) The authors carried out structural analysis and MD simulation to design 24 POP variants to create a pair of bpy-Ala variants. Although all satisfied the geometric constraints in the designing process, their response to metal ions varies significantly. To provide a better guideline, additional biochemical characterization would be necessary to speculate why some variants worked but not others. Perhaps, the authors can group 24 POP variants into four categories, one showing metal- dependent activity as expected, inverse metal-dependent activity, no metal-dependence with catalytic activity, and no metal-dependence/activity. Then,

Point-by-point Response to Reviewers – Zubi et al.

the authors may characterize any representative variants from each group to explain their distinct behavior on metal ions.

We have grouped these variants as suggested into general categories (Active/Inactive, and Inhibited/Activated/Non-responsive to Ni) along with information about the distance between the two BpyAla residues in the Closed conformation as well as the change in distance between the residues from the Open to Closed conformations (see below in response to point number 10). We are unable to notice any obvious trends that can explain why certain sites afford better switching activity compared to others. The only thing that was apparent from this presentation of the data was that mutations from H442 and I514 were not tolerated well and led to loss of activity. Perhaps, geometric information (e.g. orientation of the BpyAla residues) could reveal trends in performance, however, we did not include these constraints in our selection of sites for BpyAla incorporation, but are interested in pursuing in further studies with similar systems.

3) 167/513 and 170/513 show inverse metal-dependent activity in that they exhibit even accelerated catalytic activity upon the addition of metal ions. How is it even possible?

We, too, have wondered about this, and thank the reviewer for the opportunity to speculate on the nature of this unexpected finding. Indeed, we were surprised to find that POP_{167/513} and POP_{170/513} were activated in response to metal, rather than inhibited. This result was independently seen within two separate laboratories, confirming the activation. We conducted kinetic assays on POP_{167/513} which show improvements in both K_M and k_{cat} in the presence of Ni (II) (Supplementary Figure 5 and Supplementary Table 2). This suggests that the Ni (II) ion may play a role in stabilizing the active site (thus increasing k_{cat}) and in improving binding of the substrate to the active site (thus decreasing K_M).

4) To my best knowledge, POP is a dimeric protein. Therefore, the addition of bpy-Ala residue at one position of a protomer would form not only M(bpy)₂ moiety but also two distantly located bpy residues. Although it would influence both structure and function of the proteins, the authors neither described nor discussed this aspect in this manuscript. Because these residues are exposed to the protein surface, the latter bpy-Ala may form intermolecular complexes, possibly generating undesirable high-ordered species. Is it why the SDS-PAGE analysis was carried out in the presence of excess EDTA? Perhaps, the formation of high-ordered species might be related to their distinct response to the metal ion. Do authors examine the oligomeric structures of the proteins upon the addition of metal ions? Perhaps, size exclusion chromatography or analytical ultracentrifugation in the presence of Ni can be carried out to quantify the fraction of catalytically active and metal-responsive species.

We thank the reviewer for this question. *Pfu* POP is a monomeric protein with two domains, a β -propeller domain which restricts substrate access to the active site and a peptidase domain which contains the catalytic residues. We apologize for the confusion regarding the conditions under which SDS-PAGE analysis was performed under. The SDS-PAGE data presented in the SI was obtained with protein that had been purified by SEC in the presence of EDTA; the samples were subsequently buffer-exchanged into water prior to SDS-PAGE analysis and gels were not run in the presence of metal chelator.

We have not run size exclusion chromatography in the presence of divalent metals before, but certainly, that could be informative in future studies focused on structural characterization of these BpyAla enzyme switches. We have, however, run SEC in the absence of EDTA with the Fe-bound protein isolated by NiNTA, and the chromatograms were nearly identical in nature

Point-by-point Response to Reviewers – Zubi et al.

compared to those run after treatment with EDTA/phenanthroline or those eluted in the presence of a chelator. Given that the proteins purified by SEC without chelator retained their metal, apparent by their pink color, coupled with the similarities between chromatograms from SEC run with/without chelator, would suggest similar assembly states of the apo and metalloproteins. We do not rule out the possibility that higher-ordered species could form, however to date, we have no evidence for the formation of higher-ordered species as the reviewer has suggested. We would therefore prefer not to speculate on potential effects of their formation.

5) If the intermolecular complexation occurs, as shown below, what are the steady-state kinetic parameters of the 1:1 mixture of the 167 and 517 variants?

We thank the reviewer for this question, and as such have performed the suggested experiments. Like the kinetics of POP₁₆₇ and POP₅₁₇, the kinetics of a 1:1 mixture of POP₁₆₇ and POP₅₁₇ showed little inhibition in the presence of Ni(II), suggesting that the metal-dependent inhibition of POP_{167/517} is due to an intramolecular interaction. We have included these data in SI Figure 5 and SI Table 2 as shown above in response to reviewer 1:

^a Variant	K_M (μM)		k_{cat} (sec^{-1})	
	+ EDTA	+ NiCl ₂	+ EDTA	+ NiCl ₂
(1:1) POP ₁₆₇ and POP ₅₁₇	340 ± 19	317 ± 22	275 ± 7	225 ± 7

^aReactions were conducted in triplicate using 0-1 mM Z-Ala-Pro-pNA and 20-21 nM enzyme in 10% v/v DMSO/30 mM HEPES (pH 7.4) containing 0.8 M NaCl at 85 °C for 1 minute. Rates were determined by changes in absorbance over time at 410 nm using a calculated molar extinction coefficient for pNA (7,126 M⁻¹ cm⁻¹). Kinetic parameters were determined by the non-linear regression function in OriginPro using the Michaelis-Menten equation (Equation 2).

Additionally, we have added the following line to the end of paragraph one in the section titled *Functional characterization of a BpyAla POP Switch*:

A 1:1 mixture of POP₁₆₇ and POP₅₁₇ behaved similarly to the individual single-point variants (Supplementary Fig. 5), consistent with the proposed intra-protein cross-link rather than potential inter-protein interactions that could also result in inhibition.

Point-by-point Response to Reviewers – Zubi et al.

6) *The authors characterized one of the POP variants by UV-Vis and CD spectroscopy, demonstrating that Fe would be a surrogate metal element to show Ni-dependent chemical switch. However, neither ferric nor ferrous ion was not applied for this system in Figure 3 for no apparent reason. Please provide the catalytic activities of POP variants with Fe ion.*

Reviewer 1 also raised this important concern. We have included data for activity in the presence of Fe(II) in Figure 3b and Supplementary Figures 3 and 4. Please see response above.

7) *Please include why the authors used a cell-free expression system instead of standard e coli heterologous expression. Is it to increase protein yields of having two bpy-Ala residues in a protomer? More detailed information in the experimental setup might be necessary for readers.*

In this work, we decided to use a cell-free expression system for several reasons. Firstly, as the reviewer identifies, CFPS is volumetrically efficient and rapid for BpyAla incorporation, as ~ 1,500 µg 2TAG-sfGFP / mL of CFPS reaction can be obtained overnight. Secondly, the CFPS reaction is an open reaction environment, allowing for enzymatic assays to be conducted within the enzyme-enriched CFPS reaction by simply adding appropriate buffers and substrates without concern for factors such as membrane permeability. This importantly avoid tedious processing steps necessary for *in vivo* expression in *E. coli* for similar lysate-based enzyme assays, such as washing, centrifuging, and lysing, which must be repeated for every enzyme variant and thus becomes quickly untenable.

8) *Among the divalent metal ions that the authors used, the cupric ion is redox-active under physiological conditions, and reduction may lead to significant changes in the geometry of M(bpy)₂ and subsequent metal-dependent catalytic activities. Because the catalytic activities of POP are independent of dioxygen, the function of the metal-dependent switch can be monitored under anaerobic conditions. Can authors explore whether these systems are redox-dependent?*

We agree this could be an interesting future direction. At this point, however, we would prefer to avoid this additional complexity.

9) *Do authors only consider the distance between two residues for MD simulation? What about the directionality of the side-chains? Now that the authors have both simulation and experimental data sets, can any conclusion or criteria in the determination of two bpy-positions can be drawn and reassessed?*

This is also an excellent suggestion. We did not consider coordination geometry in our original designs, and that could certainly explain differences that we see between different divalent metals. Ongoing studies are focused on obtaining crystal structures of different LG variants with different divalent metals. We think this will be essentially to understanding the extent to which the metal ion and the protein scaffold dictate the observed coordination geometry and thus the extent to which these considerations are important for LG design. Further analysis of our simulation data could also reveal differences in the effects of side chain orientation on switch behavior. Particularly if coupled with analysis of improved variants obtained via directed evolution, this approach could provide a wealth of information regarding the possibility of rational switch improvement.

10) *Figure 2C has no information related to residues. Please provide the residue labels. In addition, add the distance between i and j residues in Supplementary Table 2. Please also include a column showing whether the pairs are responsive to metal ions as expected or not.*

Point-by-point Response to Reviewers – Zubi et al.

We have updated Figure 2C as requested.

Additionally, we have added the requested metrics to Supplementary Table 1 (previously Supplementary Table 2) as shown below:

Supplementary Table 1. Residues chosen for mutation to BpyAla in Pfu POP.

Res. #	Res. i	Res. #	Res. j	Distance i-j Closed Conformation (Å)	Delta Open-Closed (Å)	Ni ²⁺ Responsive ^a ?	Active ^b ?
216	GLN	516	SER	11.0	7.3	No	No
158	ARG	512	LEU	9.9	6.1	Inhibited	No
156	PHE	513	TYR	11.1	6.1	Inhibited	Yes
169	PRO	512	LEU	11.1	6.0	Inhibited	Yes
170	ALA	513	TYR	11.3	5.9	Activated	No
191	SER	513	TYR	10.9	5.5	Activated	No
215	ASN	524	ASN	11.6	4.4	No	Yes
214	TRP	442	HSE	11.1	-3.2	Inhibited	Complete

Point-by-point Response to Reviewers – Zubi et al.

							loss ^c
169	PRO	516	SER	9.8	-2.6	Inhibited	No
167	ASN	513	TYR	12.0	2.5	No	No
156	PHE	516	SER	11.9	2.3	Inhibited	No
158	ARG	514	ILE	9.7	-2.2	No	Complete loss ^c
159	LYS	512	LEU	11.5	1.8	No	Yes
169	PRO	510	HSE	12.5	1.8	Inhibited	Yes
169	PRO	514	ILE	10.6	-1.7	Inhibited	Complete loss ^c
159	LYS	513	TYR	10.1	-1.2	No	No
237	SER	439	GLU	11.8	-0.8	No	Yes
215	ASN	520	PRO	9.7	-0.7	No	Complete loss ^c
238	VAL	442	HSE	11.3	-0.6	No	Complete loss ^c
238	VAL	516	SER	11.6	10.6	No	No
167	ASN	516	SER	7.6	10	No	No
191	SER	520	GLU	10.7	9.2	No	Yes
167	ASN	517	VAL	11.2	10.7	Inhibited	Yes
161	LYS	516	SER	12.3	8.7	No	No
191	SER	517	VAL	9.7	9.1	Inhibited	No
159	LYS	517	VAL	10.5	9.5	Inhibited	Yes
214	TRP	524	ASN	11.5	4.2	Inhibited	Yes

^a Defined as > 50% inhibition or activation at the highest concentration of metal tested compared to apo activity. ^b Defined as having > 50% WT activity ^c Complete loss indicates < 10% WT activity

11) The metal-dependent chemical switch can be valuable, but it would not be applicable for every enzyme system. The limitation in the scope of the enzyme should be discussed. For example, metalloenzymes, having a metal cofactor are unlikely to be applicable for this chemical switch because it can be removed by EDTA treatment. In addition, only the enzymes having sufficient conformational changes at the region of substrate access and product release step can be applied.

Reviewer 1 also raised this important concern. Please see response above.

Point-by-point Response to Reviewers – Zubi et al.

12) *There is no standard deviation or experimental error in Table 1 and Supplementary Table 6. Please provide the values.*

Corrected as requested.

13) *Although $M(bpy)_2$ binding constants are fairly high, the authors always used metal ions in great excess. Is there any reason for using excess metal elements? For reversibility experiments, it would be better to use the minimum amounts of the metal element. Did the authors observe any variations in the binding affinity depending on the variants?*

We used excess metal elements in steady-state kinetic assays to ensure saturating concentrations were reached and that most enzyme was in a metal-bound state. As we observed minimal inhibition of the WT enzyme, there was no harm in using excess amounts for this purpose.

We agree with you regarding the use of minimal amounts of metal in the reversibility experiment. We tried to use lower concentrations of metals when first optimizing this experiment, however at the high temperatures which were used for kinetic assays, we observed incomplete inhibition of the enzyme, so the concentration was increased until similar levels of inhibition compared to steady-state experiments were observed.

Screening data was consistent with different binding affinities among the BpyAla variants, as variable amounts of metal were required to reach full inhibition depending on the variant and metal identity (see Supplementary Figure 3). We are interested in characterizing these affinities more rigorously to accurately compare K_d values and are currently pursuing this, however this would be featured with other experiments focused on better understanding *why* certain switches performed better than others and how to better improve our site-selection criteria (e.g. microscopy to look at distribution of conformations, crystallography, FRET, etc.).

Reviewers' Comments:

Reviewer #1:

Remarks to the Author:

I congratulate the authors to this successfully revised version of their manuscript. I like to thank the authors for their meticulous efforts to answer my questions and incorporate my suggestions. All of my concerns have been taken care of and the updated article is nice to read, consistent and tells a really exciting story.

As a result of rereading the manuscript, I have only some further minor points for improvement:

- The story suffers from a missing description of the design principle of BpyAla. Although it is shown in Figure 1a, it is not explained in text form neither in the main text nor in the figure caption. The authors should supplement such a description in the caption to Figure 1 and at least mention the design principle in the first chapter of the results in the first or at the beginning of the second paragraph. Otherwise, the reader will not understand why double variants are required for the design strategy (p. 4, l.142). Similarly, Figure 1b would benefit from a detailed description in the caption.

- P.1, l. 46: I believe "however" should be positioned after "... in a stimulus-dependent manner"

- P.2, l.58: LOV stands for "light-oxygen-voltage" and not "light, oxygen and voltage"

- P. 2, l. 63: It should be "*Arabidopsis thaliana*" (in italic)

- P.4, l.145 "... that undergoes catalytically relevant conformation changes." This is too vague.

Many different kinds of conformational changes exist and I believe not all of them can be manipulated with the authors' technique. I suggest to specify (e.g. "relevant conformational domain change").

- P.7, l. 238 (Supplementary Fig. 6 related): What is the authors' definition of "minimal"? Please specify using approximations to avoid misunderstanding, e.g.:

o Wt ~95% pure (ok)

o POP167 ~90% pure and <5% truncation product (ok)

o POP517 ~85% pure and ~10% truncation product (still ok)

o POP167/517 ~90% pure and <5% truncation product (ok)

o POP159/517, POP169/510, POP169/512, POP167/513 ~60% pure and 30-40% truncation product ◊ in my opinion not "minimal"; they authors might want to add their explanation given in the rebuttal that truncated POP shows affinity for Ni-NTA

- P.7, l. 241, Supplementary Table 2: Since the enzymes are not homogenous (purity ~60%), I suggest to give v_{max} instead of k_{cat} as $k_{cat} = v_{max}/[E]$ and $[E]$ is not known due to impurity (In comparison, k_{cat} in Table 1 is fine since the variants are >85% pure) ◊ or give in general v_{max} to stay consistent

- P.7, l.243: k_{cat} and K_m could not be measured in presence of Ni(II) ◊ for quantitative comparison, the authors could compare k_{cat}/K_m as here they only need to fit the initial linear part of the Michaelis-Menten curve

Reviewer #2:

Remarks to the Author:

The authors addressed all essential concerns and questions and revised the manuscript accordingly. Although the strategy developed herein as "Metal-Responsive Regulation of Enzymes" can be applied for specific systems for now, and thus it needs further improvements for in vivo experiments or else, the proof of concept is noteworthy, and they demonstrated the experiments thoroughly. Therefore, I believe that it is suitable for publishing in Nature Communications.

Reviewer #1 (Remarks to the Author):

- The story suffers from a missing description of the design principle of BpyAla. Although it is shown in Figure 1a, it is not explained in text form neither in the main text nor in the figure caption. The authors should supplement such a description in the caption to Figure 1 and at least mention the design principle in the first chapter of the results in the first or at the beginning of the second paragraph. Otherwise, the reader will not understand why double variants are required for the design strategy (p. 4, l.142). Similarly, Figure 1b would benefit from a detailed description in the caption.

The manuscript notes that the Bpy LG works via bis-bidentate metal binding on P2, Line 91:

Initial efforts focused on bipyridine (Bpy) LGs that could induce reversible activation of enzymes in the presence/absence of metal salts (Figure 1a). Specifically, bis-bidentate metal binding by the Bpy LGs would restrict the enzyme to a closed conformation incapable of turnover while removal of the metal would allow the enzyme to access conformational changes required for turnover.

To make this more clear in Figure 1, we modified the figure description as follows:

(a) Reversible metal regulation of *Pfu* prolyl oligopeptidase (POP) via bis-bidentate metal binding by a BpyAla LG pair.

- P.7, l. 238 (Supplementary Fig. 6 related): What is the authors' definition of "minimal"? Please specify using approximations to avoid misunderstanding, e.g.:

- o Wt ~95% pure (ok)
- o POP167 ~90% pure and <5% truncation product (ok)
- o POP517 ~85% pure and ~10% truncation product (still ok)
- o POP167/517 ~90% pure and <5% truncation product (ok)
- o POP159/517, POP169/510, POP169/512, POP167/513 ~60% pure and 30-40% truncation product \diamond in my opinion not "minimal"; they authors might want to add their explanation given in the rebuttal that truncated POP shows affinity for Ni-NTA

The reviewer is correct that some of the variants that were lumped into this general statement had impurity levels beyond what can reasonably be considered "minimal". We have rewritten this passage to state the full range of purity observed and to specify the high purity of the key variants as follows:

Purified variants were produced in yields of approximately 50 mg/L, and BpyAla incorporation was confirmed by intact protein ESI-MS. The isolated variants included ~5-40% of POP that had been truncated at the first amber stop codon due to the inherent affinity of POP for Ni-NTA resin, though >90% purity was observed for POP_{167/517}, POP₁₆₇, and POP₅₁₇ (Supplementary Fig. 6).

The figures caption for Supplementary Fig. 6 was also updated (relevant section only):

In addition to a band corresponding to the MW of the desired full-length protein, we also observe varying amounts of the truncated protein depending on the variant (<5% - ~40%), that forms because of competition between release factor 1 mediated (RF1) termination of translation and BpyAla incorporation at the amber stop codon¹. Truncated protein was isolated along with full-length protein after IMAC due to apparent affinity of the truncated POP side products for Ni-NTA.

- P.7, l. 241, Supplementary Table 2: Since the enzymes are not homogenous (purity ~60%), I suggest to give v_{max} instead of k_{cat} as $k_{cat} = v_{max}/[E]$ and $[E]$ is not known due to impurity (In comparison, k_{cat} in Table 1 is fine since the variants are >85% pure) \diamond or give in general v_{max} to stay consistent

We added V_{max} and a note about this issue to Supplementary Table 2:

Supplementary Table 2. Steady-state kinetic parameters for selected POP variants.

^a Variant	K_M (μM)		k_{cat} (sec^{-1})		V_{max} ($\mu M/sec$) ^b	
	+ EDTA	+ $NiCl_2$	+ EDTA	+ $NiCl_2$	+ EDTA	+ $NiCl_2$
POP _{159/517}	265 ± 38	523 ± 73	139 ± 8	57 ± 4	2.95 ± 0.16	1.21 ± 0.09
POP _{169/510}	409 ± 20	NA	154 ± 4	NA	4.60 ± 0.11	NA
POP _{169/512}	428 ± 36	NA	91 ± 4	NA	0.993 ± 0.039	NA
POP _{167/513}	854 ± 102	661 ± 43	219 ± 16	259 ± 9	4.37 ± 0.31	5.18 ± 0.18
(1:1) POP ₁₆₇ and POP ₅₁₇	340 ± 19	317 ± 22	275 ± 7	225 ± 7	5.50 ± 0.13	4.49 ± 0.15

^aReactions were conducted in triplicate using 0-1 mM Z-Ala-Pro-pNA and 20-21 nM enzyme in 10% v/v DMSO/30 mM HEPES (pH 7.4) containing 0.8 M NaCl at 85 °C for 1 minute. Rates were determined by changes in absorbance over time at 410 nm using a calculated molar extinction coefficient for pNA ($7,126 M^{-1} cm^{-1}$). Kinetic parameters were determined by the non-linear regression function in OriginPro using the Michaelis-Menten equation (Equation 2). ^b V_{max} is also provided due to the variable purity of the variants specified used here (see Supplementary Figure 6 below).

- P.7, l.243: k_{cat} and K_M could not be measured in presence of Ni(II) \diamond for quantitative comparison, the authors could compare k_{cat}/K_M as here they only need to fit the initial linear part of the Michaelis-Menten curve

We would prefer not to compare k_{cat}/K_M values determined from proper fitting of saturation kinetics vs initial part of such data.

- P.1, l. 46: I believe “however” should be positioned after “... in a stimulus-dependent manner”
- P.2, l.58: LOV stands for “light-oxygen-voltage” and not “light, oxygen and voltage”
- P. 2, l. 63: It should be “*Arabidopsis thaliana*” (in italic)
- P.4, l.145 “... that undergoes catalytically relevant conformation changes.” This is too vague. Many different kinds of conformational changes exist and I believe not all of them can be manipulated with the authors’ technique. I suggest to specify (e.g. “relevant conformational domain change”).

All corrected as requested.